# Joint Modeling of fMRI and EEG Imaging Using Ordinary Differential Equation-Based Hypergraph Neural Networks

**Yan Zhang, Yang Gao, Min Li**[*]
School of Computer Science and Engineering
Central South University
Changsha, Hunan 410083
`limin@mail.csu.edu.cn`

## Abstract

Fusing multimodal brain imaging has been a hot topic since different modalities of brain imaging can provide complementary information. However, due to the size of simultaneous recorded fMRI-EEG dataset being limited and the substantial discrepancy between hemodynamic responses of fMRI and neural oscillations of EEG, the joint modeling of fMRI and EEG images is a rarely explored area and has not yielded satisfactory results. Existing studies have also indicated that the relationships between region of interest (ROI) are not one-to-one when synchronizing fMRI and EEG. Current graph-based multimodal modeling methods overlook those information. Based on this, we propose a hypergraph based fMRI-EEG modeling framework for asynchronous fMRI-EEG data named FE-NET. To the best of our knowledge, this is the first attempt to jointly model asynchronous EEG and fMRI data as Neural ODEs based hypergraph. Extensive experiments have demonstrated that the proposed FE-NET outperforms many state-of-the-art brain imaging modeling methods. Meanwhile, compared to simultaneously recorded fMRI-EEG data, asynchronously acquired fMRI-EEG data is less costly, which demonstrates the practical applicability of our method.

## 1 Introduction

A major goal in neuroimaging research is to develop predictive models aimed at analyzing the association between comprehensive functional connectivity patterns across the brain and behavioral traits, which can enhance the understanding of both normal brain function and dysfunction [1]. A robust modeling method plays a significant role in assisting the diagnosis of neurological disorders, predicting infant neurodevelopment, and assessing sleep quality [2; 3; 4]. Usually, researchers model brain imaging in the form of graphs to accomplish this goal [5].

In recent years, modeling multimodal brain imaging has emerged as a focal point of research. fMRI indirectly measures brain activity through the Blood Oxygen Level Dependent (BOLD) signal, while EEG directly records brain activity by measuring the electric fields generated by cortical pyramidal neurons. EEG enables non-invasive observation of brain electrical activity and serves as a complementary modality to fMRI:

The integration of fMRI with EEG signals offers a promising approach to bridging the gap between hemodynamic responses and neural oscillations, providing insights into the origins of fMRI signals and their large-scale patterns. However, deep learning-based joint modeling of asynchronously recorded fMRI and EEG is a relatively underexplored area and has not yielded satisfactory results.

---

[*]Corresponding author.

39th Conference on Neural Information Processing Systems (NeurIPS 2025).

The reasons for this issue are twofold. One reason lies in the incompatibility of the equipment, the sample size of simultaneously recorded fMRI-EGG datasets is markedly smaller than that of asynchronously acquired datasets, thereby limiting the reliability of extant research relying on synchronous recordings, another reason is the fundamental differences of the signal type make the data distribution between fMRI and EEG highly complex, thereby rendering it difficult to bridge the gap between them.

Existing studies [6] have indicated that the relationships between regions of interest (ROI) are not one-to-one when synchronizing fMRI and EEG, Hyperedges in hypergraphs can connect multiple nodes, providing a more comprehensive description of this relationship.

Neural ordinary differential equations (Neural ODEs) [7] can dynamically propagate information across arbitrary timestamps, accommodating mismatched sampling rates and asynchronous time points without manual alignment, the potential of using Neural ODEs for modeling asynchronous data has been widely reported [8].

Based on the existing issues, we propose a fMRI-EEG modeling method based on Neural ODEs hypergraphs. Our main innovations are as follows:

1) We have designed a GAN-based fMRI-EEG hypergraph generation (FEH) module, which employs a novel Optimal fMRI-EEG Isomorph algorithm (OFEI) and Interactive Hyperedge Neurons (IHEN) to generate multiple hypergraphs at various temporal and spatial scales. The strength of GAN-based FEH lies in their ability to learn the complex data distribution between fMRI and EEG through synchronized operation of OFEI and IHEN, thus narrowing the gap between the hemodynamic responses and the neural oscillations.

2) We proposed a novel dynamic fMRI-EEG hypergraph embedding module (FED), which leverages Neural ODEs to capture latent temporal dependencies between fMRI's slow hemodynamic changes (s̃econds) and EEG's rapid neural oscillations (m̃illiseconds), Neural ODEs naturally handles inconsistency sampling rates and temporal mismatches by learning differential equations governing system dynamics for asynchronous data.

3) To the best of our knowledge, this is the first attempt to model asynchronous EEG and fMRI data as Neural ODEs based hypergraph. Comprehensive experiments show that the performance of the proposed FE-NET surpasses many state-of-the-art methods, compared to simultaneously recorded fMRI-EEG data. Asynchronously acquired fMRI-EEG data is less costly, which demonstrates the practical applicability of our method.

## 2 Related Work

### 2.1 Hypergraph Learning

In recent years, the efficacy of hypergraphs in modeling and comprehending intricate correlations has become evident. Initially introduced by [9], hypergraph learning embodies a transductive learning approach, conceptualized as a dissemination process within the hypergraph structure.

Motivated by the remarkable success of deep learning, some researchers have ventured into developing deep hypergraph learning techniques. For instance, Feng et al. [10] introduced hypergraph neural networks (HGNN) to model and understand beyond-pairwise intricate correlations. Unlike Graph Neural Networks (GNN), HGNN devises a vertex-hyperedge-vertex information propagation schema to iteratively glean data representation. Bai et al. [11] delved deeper into the attention mechanism on hypergraphs, aligning with the hypergraph convolution paradigms delineated in HGNN. Inspired by [12], Hyper-Atten introduces a hyperedge-vertex attention learning module to dynamically discern the importance of different vertices within the same hyperedge, thereby unveiling intrinsic correlations. Moreover, Yadati et al. [13] proposed HyperGCN, a method for training a Graph Convolutional Network (GCN) on hypergraphs for semi-supervised learning. HyperGCN is structured based on the spectral theory of hypergraphs. Initially, HyperGCN transforms a hypergraph into a simple weighted graph via a specific strategy and subsequently executes standard GCN on the graph to glean data representations.

As elucidated, most prevailing deep hypergraph learning methods stem from the spectral theory of hypergraphs. Consequently, these methods are typically defined on static hypergraphs system, constraining their applications. Failing to capture the continuous evolution of fMRI-EEG hypergraph

representations. Many important information in the dynamic continuous representations between fMRI and EEG has been lost.

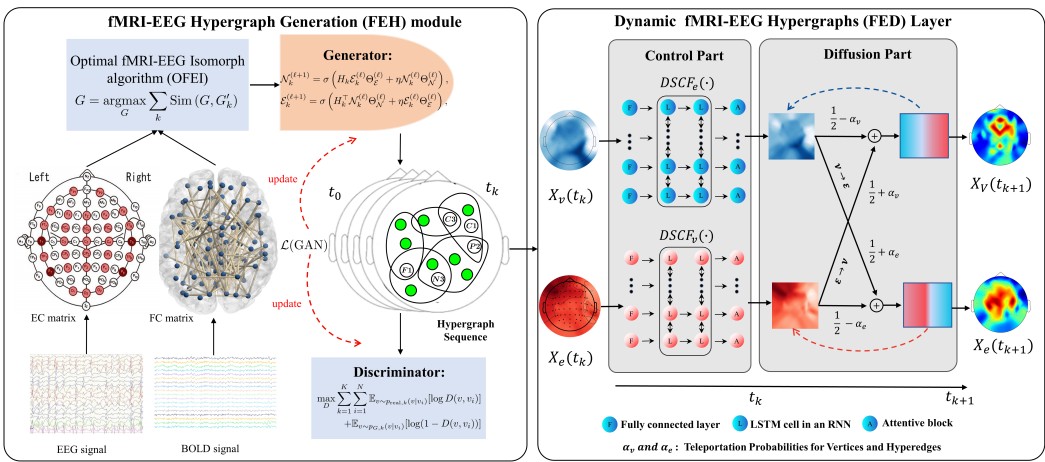

Figure 1: Overall process of GAN-based hypergraph generation (FEH) module and Dynamic hypergraphs embedding (FED) module.

# 3 Method

As shown in Figure 1, our method consists of two steps. Firstly, multiple fMRI-EEG hypergraphs are generated at various temporal and spatial scales by GAN-based fMRI-EEG hypergraph generation module. Then, these hypergraphs are inputted into the dynamic fMRI-EEG hypergraph embedding module.

## 3.1 Preliminary

**Notations.** In the hypergraph, each hyperedge, which is a subset of the vertex set, exhibits a distinction from the simple graph. Typically, a hypergraph is defined as $\mathcal{G} = (\mathcal{V}, \mathcal{E})$, where $\mathcal{V}$ and $\mathcal{E}$ denote the vertex set and hyperedge set, respectively. The hyperedges are represented by an incidence matrix $\boldsymbol{H} \in \{0,1\}^{|\mathcal{V}| \times |\mathcal{E}|}$, with its elements defined as $H_{v,e} = \mathbf{1}(v \in e)$ utilizing the indicator function $\mathbf{1}(\cdot)$. The degree of a vertex $v \in \mathcal{V}$ is determined by the summation of the hyperedge occurrences $H_{v,e}$ over the hyperedge set $\mathcal{E}$, denoted as $d(v) = \sum_{e \in \mathcal{E}} H_{v,e}$. Similarly, the degree of a hyperedge $e \in \mathcal{E}$ is determined by the summation of the vertex occurrences $H_{v,e}$ over the vertex set $\mathcal{V}$, denoted as $\delta(e) = \sum_{v \in \mathcal{V}} H_{v,e}$. The diagonal degree matrices for vertices and hyperedges are represented as $\boldsymbol{D}_v = \mathrm{diag}(\boldsymbol{d})$ and $\boldsymbol{D}_e = \mathrm{diag}(\boldsymbol{\delta})$, respectively.

**Hypergraph neural networks.** The majority of contemporary hypergraph neural networks adhere to the message-passing framework. Within each layer, the input vertex representations are initially aggregated into hyperedges, and subsequently, the output vertex representations are derived from the corresponding hyperedges. Formally, in the $k$-th layer, the output vertex representations $\boldsymbol{x}_v^{(k)}$ are computed based on the previous representations through the employment of the aggregation function AGG and the update function UPD as follows:

$$\boldsymbol{x}_v^{(k)} = \mathrm{UPD}\left(\boldsymbol{x}_v^{(k-1)}, \mathrm{AGG}\left(\left\{\boldsymbol{x}_e^{(k)} : e \in \mathcal{N}_e(v)\right\}\right)\right), \quad \boldsymbol{x}_e^{(k)} = \mathrm{AGG}\left(\left\{\boldsymbol{x}_v^{(k-1)} : v \in \mathcal{N}_v(e)\right\}\right) \tag{1}$$

where $\mathcal{N}_e(v)$ and $\mathcal{N}_v(e)$ correspond to the vertex and hyperedge neighbor functions, respectively.

## 3.2 GAN-based fMRI-EEG hypergraph generation (FEH) module

### 3.2.1 Optimal fMRI-EEG Isomorph algorithm (OFEI)

As shown in Figure 1 (left), each region of interest (ROI) is conceptualized as a node within the framework. Initially, the Dynamic Hypergraph Construction algorithm (DHC) defined in [14], is

employed to compute the initial hypergraphs $\{G'_k\}_{k=1}^t$ for each subject within the dataset, where $t$ represents the aggregate number of samples. If these initial hypergraphs $\{G'_k\}$ are directly utilized in the generative process to achieve multimodal connectivity, the robustness of the resultant generation may be compromised. In response to this limitation, we design a novel Optimal fMRI-EEG Isomorph algorithm (OFEI), which is capable of calculating a hypergraph $G$ that exhibits optimal isomorphism relative to the initial hypergraphs $\{G'_k\}$,

Let $G = (V, E)$ and $G' = (V, E')$ share the same vertex set and have the same number of hyperedges, $m = |E| = |E'|$. Index the edges as $E = \{e_1, \ldots, e_m\}$ and $E' = \{e'_1, \ldots, e'_m\}$. For an edge $e$, let $\mathcal{V}_G(e) \subseteq V$ denote its incident-vertex set. Define the edge-wise Jaccard similarity

$$s(e, e') = \frac{|\mathcal{V}_G(e) \cap \mathcal{V}_{G'}(e')|}{|\mathcal{V}_G(e) \cup \mathcal{V}_{G'}(e')|}. \tag{2}$$

The homogeneity between $G$ and $G'$ is

$$\text{Sim}(G, G') := \max_{\sigma \in S_m} \frac{1}{m} \sum_{i=1}^m s\big(e_i, e'_{\sigma(i)}\big), \tag{3}$$

i.e., the maximum over bijections $f : E \to E'$ (since $|E| = |E'|$).

Writing $\mathcal{H}(V, m)$ for the class of hypergraphs on $V$ with $m$ hyperedges, OFEI solves

$$G^* \in \arg \max_{G \in \mathcal{H}(V,m)} \sum_{k=1}^t \text{Sim}\big(G, G'_k\big). \tag{4}$$

With the optimal isomorph $G^*$ in hand, we construct subject-specific hypergraphs by concatenating $G^*$ with each $G'_k$,

$$G_k := G^* \,\|\, G'_k, \tag{5}$$

where $\|$ denotes edgewise concatenation on the common vertex set (allowing multiplicities if applicable). The corresponding incidence matrices are then obtained as

$$H_k = H(G_k) \quad \text{(equivalently, } H_k = [\, H(G^*) \,\big|\, H(G'_k) \,]\text{).} \tag{6}$$

The rationale behind OFEI lies in its ability to establish a metric for measuring the similarity among hypergraphs, thereby enhancing the correlation between fMRI and EEG data. This ensures the generated hypergraph preserves the topological consistency of brain connectivity across modalities, the data alignment and node correspondence part are provided in supplementary material.

### 3.2.2  Generator formed by interactive hyperedge neurons module

For the $k$-th participant, we initialize node representations $\mathcal{N}_k^{(0)} \in \mathbb{R}^{N \times d}$ directly from their BOLD signal matrix $B_k$, where $N$ denotes the ROI count and $d$ the temporal length. Simultaneously, hyperedge representations $\mathcal{E}_k^{(0)} \in \mathbb{R}^{m \times N}$ are constructed via $\mathcal{E}_k^{(0)} = H_k^\top S_k$, where $S_k \in \mathbb{R}^{N \times N}$ is the effective connectivity matrix, $H_k \in \mathbb{R}^{N \times m}$ is the incidence structure, and $m$ represents the hyperedge cardinality.

The generative architecture $\mathcal{G}$ consists of $L$ cascaded Interactive Hyperedge Neuron (IHEN) transformations. For layer index $\ell \in \{0, 1, \ldots, L-1\}$, the IHEN propagation rules are:

$$\mathcal{N}_k^{(\ell+1)} = \sigma \left( H_k \mathcal{E}_k^{(\ell)} \Theta_{\mathcal{E}}^{(\ell)} + \eta \mathcal{N}_k^{(\ell)} \Theta_{\mathcal{N}}^{(\ell)} \right), \mathcal{E}_k^{(\ell+1)} = \sigma \left( H_k^\top \mathcal{N}_k^{(\ell)} \Theta_{\mathcal{N}}^{(\ell)} + \eta \mathcal{E}_k^{(\ell)} \Theta_{\mathcal{E}}^{(\ell)} \right), \tag{7}$$

where $\Theta_{\mathcal{N}}^{(\ell)}$ and $\Theta_{\mathcal{E}}^{(\ell)}$ are learnable transformation matrices for nodes and hyperedges respectively, and $\eta$ is a mixing coefficient.

From the terminal representations $\mathcal{N}_k^{(L)}$ and $\mathcal{E}_k^{(L)}$, we compute node-to-hyperedge affinities and hyperedge importances. Define the affinity of node $v_i$ to hyperedge $e_j$ as:

$$\phi_{k,j}(i) = H_k(i, j) \cdot \langle \mathcal{N}_k^{(L)}[i, :], \mathcal{E}_k^{(L)}[j, :] \rangle, \tag{8}$$

and the importance weight of hyperedge $e_j$ as:

$$\omega_k(j) = \left\| \mathcal{E}_k^{(L)}[j, :] \right\|_2. \tag{9}$$

The multimodal connectivity tensor $\mathcal{M}_k \in \mathbb{R}^{N \times N}$ is then assembled via:

$$\mathcal{M}_k[p,q] = \sum_{j=1}^{m} \omega_k(j) \cdot \phi_{k,j}(p) \cdot \phi_{k,j}(q). \tag{10}$$

Finally, node-wise correlation scores are computed by integrating connectivity strengths with feature similarities:

$$\mathcal{C}_k(i) = \frac{1}{N} \sum_{q=1}^{N} \mathcal{M}_k[i,q] \cdot \langle \mathcal{N}_k^{(L)}[i,:], \mathcal{N}_k^{(L)}[q,:] \rangle. \tag{11}$$

### 3.2.3 Discriminator and Loss Function

The discriminator $\mathcal{D}$ utilizes a standard multilayer perceptron architecture. Given an arbitrary initial node $u_0$, we initiate random walks from this origin. For a walk of length $T$ terminating at $u_T$, the traversal sequence $\mathbf{u} = (u_0, u_1, \ldots, u_{T-1}, u_T, u_{T+1})$ satisfies the boundary condition $u_{T+1} = u_{T-1}$, establishing $u_T$ as the terminal node.

For the functional connectivity matrix $\mathbf{F}_k$ of subject $k$, the probability of realizing trajectory $\mathbf{u}$ is:

$$P_k^{\text{emp}}(\mathbf{u}) = \prod_{i=1}^{T} \frac{\mathbf{F}_k(u_{i-1}, u_i)}{\sum_j \mathbf{F}_k(u_{i-1}, j)} \times \frac{\mathbf{F}_k(u_T, u_{T-1})}{\sum_j \mathbf{F}_k(u_T, j)} \tag{12}$$

Denoting $\Omega(u_0 \to v)$ as the collection of all paths from $u_0$ to destination $v$, the marginal probability is:

$$P_k^{\text{emp}}(v \mid u_0) = \sum_{\gamma \in \Omega(u_0 \to v)} P_k^{\text{emp}}(\gamma) \tag{13}$$

For the generated multimodal connectivity matrix $\mathbf{M}_k$, the corresponding trajectory probability becomes:

$$P_k^{\text{gen}}(\mathbf{u}) = \prod_{i=1}^{T} \frac{\mathbf{M}_k(u_{i-1}, u_i)}{\sum_j \mathbf{M}_k(u_{i-1}, j)} \times \frac{\mathbf{M}_k(u_T, u_{T-1})}{\sum_j \mathbf{M}_k(u_T, j)} \tag{14}$$

with marginal distribution:

$$P_k^{\text{gen}}(v \mid u_0) = \sum_{\gamma \in \Omega(u_0 \to v)} P_k^{\text{gen}}(\gamma) \tag{15}$$

The discriminator objective function is formulated as:

$$\mathcal{L}_\mathcal{D} = \max_{\mathcal{D}} \sum_{k=1}^{K} \sum_{n=1}^{N} \left[ \mathbb{E}_{v \sim P_k^{\text{emp}}(\cdot | u_n)}[\log \mathcal{D}(v, u_n)] + \mathbb{E}_{v \sim P_k^{\text{gen}}(\cdot | u_n)}[\log(1 - \mathcal{D}(v, u_n))] \right] \tag{16}$$

where $K$ represents the cohort size and $N$ the number of nodes.

The generator objective, driven by discriminator feedback, is:

$$\mathcal{L}_\mathcal{G} = \max_{\mathcal{G}} \sum_{k=1}^{K} \sum_{n=1}^{N} \mathbb{E}_{v \sim P_k^{\text{gen}}(\cdot | u_n)}[\log \mathcal{D}(v, u_n)] \tag{17}$$

The OFEI framework achieves optimal structural correspondence across multimodal fMRI-EEG hypergraph representations, while the IHEN mechanism dynamically modulates hyperedge weights $\phi_{k,j}(i)$ through learned node-hyperedge affinities, emphasizing neurobiologically significant connectivity patterns (e.g., functionally coherent brain networks).

The FEH module combines **structural optimization** (OFEI), **dynamic feature fusion** (IHEN), and **distribution alignment** (GAN) to generate hypergraphs that bridge fMRI and EEG's gaps at various temporal and spatial scales, thus can narrow the gap between the hemodynamic responses and the neural oscillations, subsequently, fMRI-EEG hypergraphs will be sent into the dynamic FMRI-EEG hypergraphs embedding module for feature extraction.

## 3.3 Dynamic fMRI-EEG hypergraphs embedding (FED) module

In this section, we first give the definition of dynamic fMRI-EEG hypergraph systems and provide a specific dynamic fMRI-EEG hypergraph systems based on ODE. Next, we describe the detailed neural implementation of the FED $^{\text{ode}}$ framework, the overall process of FED module is shown in Figure 1.

### 3.3.1 Definition of dynamic fMRI-EEG hypergraph systems

Given a hypergraph $\mathcal{G}$, a corresponding vertex feature matrix $\boldsymbol{Z}_v \in \mathbb{R}^{|\mathcal{V}| \times c}$, and a corresponding hyperedge feature matrix $Z_e \in \mathbb{R}^{|\mathcal{E}| \times c}$, our goal is to learn a vertex representation $\boldsymbol{Y}_v$ and a hyperedge representation $\boldsymbol{Y}_e$. We first define dynamic fMRI-EEG hypergraph systems based on the following equation:

$$\left[ \begin{array}{c} \dot{\boldsymbol{X}}_v \\ \dot{\boldsymbol{X}}_e \end{array} \right] = f \left( \left[ \begin{array}{c} \boldsymbol{X}_v(t) \\ \boldsymbol{X}_e(t) \end{array} \right] \right) \text{ and } \left[ \begin{array}{c} \boldsymbol{X}_v(0) \\ \boldsymbol{X}_e(0) \end{array} \right] = \left[ \begin{array}{c} \boldsymbol{Z}_v \\ \boldsymbol{Z}_e \end{array} \right] \tag{18}$$

Here, $\boldsymbol{X}_v(t)$ and $\boldsymbol{X}_e(t)$ denote the matrices representing vertex and hyperedge features at time $t$, respectively. The function $f$ encapsulates the rate of change within the dynamic system, with $\boldsymbol{Z}_v$ and $\boldsymbol{Z}_e$ representing the initial conditions for vertex and hyperedge features, respectively. The persistence of timestamp $t$ allows the hypergraph dynamic systems to continuously update representation states.

**ODE-based hypergraph dynamic system.** The velocity function $f$ in Equation(18) may be characterized variably. It is proposed to interpret the velocity function as a composite of a control function and a diffusion function, thereby formulating an ODE-based hypergraph dynamic system in the following manner:

$$\left[ \begin{array}{c} \dot{\boldsymbol{X}}_v \\ \dot{\boldsymbol{X}}_e \end{array} \right] = \left[ \begin{array}{c} C_v \left( \boldsymbol{X}_v(t) \right) \\ C_e \left( \boldsymbol{X}_e(t) \right) \end{array} \right] + \boldsymbol{H} \left[ \begin{array}{c} \boldsymbol{X}_v(t) \\ \boldsymbol{X}_e(t) \end{array} \right]. \tag{19}$$

In the initial segment of the equation, $C_v$ and $C_e$ serve as control functions, which correspond to the velocity of control for the representations of vertices and hyperedges, respectively. The subsequent segment represents the diffusion term.

$$\left[ \begin{array}{c} \boldsymbol{X}_v(T) \\ \boldsymbol{X}_e(T) \end{array} \right] = \left[ \begin{array}{c} \boldsymbol{X}_v(0) \\ \boldsymbol{X}_e(0) \end{array} \right] + \int_0^T C \left( \left[ \begin{array}{c} \boldsymbol{X}_v(t) \\ \boldsymbol{X}_e(t) \end{array} \right] \right) dt \tag{20}$$

Given the initial vertex features $\boldsymbol{X}_v(0)$ and hyperedge features $\boldsymbol{X}_e(0)$ as inputs, the corresponding vertex representations $\boldsymbol{X}_v(T)$ and hyperedge representations $\boldsymbol{X}_e(T)$ at time $T$ are generated through integration, as outlined in Equation(20), for learning tasks within the hypergraph.

We anticipate proposing a multi-layer neural network framework, denoted as FED$^{\text{ode}}$. Initially, we adopt the Lie-Trotter splitting method, as delineated by Geiser [15], for the discretization of Equation(19), delineated below:

$$\begin{aligned} \left[ \begin{array}{c} \boldsymbol{X}_v \left( t + \frac{1}{2} \right) \\ \boldsymbol{X}_e \left( t + \frac{1}{2} \right) \end{array} \right] &= \left[ \begin{array}{c} \boldsymbol{X}_v(t) \\ \boldsymbol{X}_e(t) \end{array} \right] + \left[ \begin{array}{c} C_v \left( \boldsymbol{X}_v(t) \right) \\ C_e \left( \boldsymbol{X}_e(t) \right) \end{array} \right], \\ \left[ \begin{array}{c} \boldsymbol{X}_v(t+1) \\ \boldsymbol{X}_e(t+1) \end{array} \right] &= \left[ \begin{array}{c} \boldsymbol{X}_v \left( t + \frac{1}{2} \right) \\ \boldsymbol{X}_e \left( t + \frac{1}{2} \right) \end{array} \right] + \boldsymbol{A} \left[ \begin{array}{c} \boldsymbol{X}_v \left( t + \frac{1}{2} \right) \\ \boldsymbol{X}_e \left( t + \frac{1}{2} \right) \end{array} \right], \end{aligned} \tag{21}$$

In the model, the time step is incorporated into control functions $C_v$ and $C_e$, as well as the diffusion matrix $\boldsymbol{A}$. Lie-Trotter discretization strategy allow separates control (modality-specific dynamics) part and diffusion (cross-modal interactions) part, allowing asynchronous updates of fMRI and EEG features while preserving their intrinsic temporal characteristics.

### 3.3.2 Neural implementation of the FED $^{\text{ode}}$ framework

we delineate the integration of the FED$^{\text{ode}}$ framework into the antecedent analysis by elucidating the neural instantiation of both the control and diffusion phases within the FED$^{\text{ode}}$ layer, separately. A graphical representation of the FED$^{\text{ode}}$ framework is provided in Figure 1 (right).

**Neural implementation of control step.** To extract spatio-temporal information from fMRI and EEG data, we propose a Dual-Stream Control Function (DSCF). As illustrated in Figure 1, DSCF incorporates components where F, L, and A represent, respectively, a fully connected layer, an LSTM

cell in an RNN, and an attentive block aligned with an RNN output defined in literature [16]. The control function of FED $^{\text{ode}}$ layer is specifically articulated through the subsequent equation:

$$
\left[ \begin{array}{c} \boldsymbol{X}_v\left(t + \frac{1}{2}\right) \\ \boldsymbol{X}_e\left(t + \frac{1}{2}\right) \end{array} \right] = \left[ \begin{array}{c} \boldsymbol{X}_v(t) \\ \boldsymbol{X}_e(t) \end{array} \right] + \left[ \begin{array}{c} DSCF_v(\boldsymbol{X}_v(t)) \\ DSCF_e(\boldsymbol{X}_e(t)) \end{array} \right] \tag{22}
$$

In this study, the temporal dynamics of brain activity can convey essential information. Therefore, using a LSTM to capture the spectral variations within the activation sequence of networks is a rational approach. The Attention Neural Network (ANN) [17] applies distinct weights to various output steps of RNNs to prioritize the most critical time periods, thereby playing a crucial role in subject-independent hypergraph encoding. Input and output parameters for FN, LSTM, and ANN are provided in the Supplementary Material.

**Neural implementation of diffusion step.** The configuration of the diffusion matrix $\boldsymbol{A}$ is paramount. Should it be improperly formulated, both vertex and hyperedge representations are prone to diverge and become intractable, delineated as follows:

$$
\left[ \begin{array}{c} \boldsymbol{X}_v(t+1) \\ \boldsymbol{X}_e(t+1) \end{array} \right] = \left[ \begin{array}{c} \frac{1}{2}\boldsymbol{X}_v\left(t + \frac{1}{2}\right) \\ \frac{1}{2}\boldsymbol{X}_e\left(t + \frac{1}{2}\right) \end{array} \right] + \boldsymbol{A} \left[ \begin{array}{c} \boldsymbol{X}_v\left(t + \frac{1}{2}\right) \\ \boldsymbol{X}_e\left(t + \frac{1}{2}\right) \end{array} \right]
$$
$$
\boldsymbol{A} = \left[ \begin{array}{cc} -\alpha_v \boldsymbol{I} & (\frac{1}{2} + \alpha_v)\boldsymbol{D}_v^{-1}\boldsymbol{H} \\ (\frac{1}{2} + \alpha_e)\boldsymbol{D}_e^{-1}\boldsymbol{H}^{\top} & -\alpha_e \boldsymbol{I} \end{array} \right], \tag{23}
$$

In the context where $\alpha_v$ and $\alpha_e$ serve as hyperparameters, these denote the teleportation probabilities associated with vertices and hyperedges, respectively. We proceed to decompose the matrix multiplication term to derive the vertex representation: $\boldsymbol{X}_v(t+1) = \left(\frac{1}{2} - \alpha_v\right)\boldsymbol{X}_v\left(t + \frac{1}{2}\right) + (\frac{1}{2} + \alpha_v)\boldsymbol{D}_v^{-1}\boldsymbol{H}\boldsymbol{X}_e\left(t + \frac{1}{2}\right)$. Here, the initial term indicates that vertex representations are maintained with a preservation factor of $\frac{1}{2} - \alpha_v$ during the diffusion process, while the latter term signifies the integration of representations from hyperedges directly linked to each vertex, weighted by $\alpha_v$. In this arrangement, $\boldsymbol{H}\boldsymbol{X}_e\left(t + \frac{1}{2}\right)$ corresponds to vertex-level aggregation, and $\boldsymbol{D}_v^{-1}$ serves as a normalization matrix for averaging.

Control part in FEH refine modality-specific features (i.e., isolating fMRI spatial patterns from EEG spectral power). The Diffusion part can propagates information across the hypergraph structure, capturing long-range dependencies (i.e., cross-modal interactions between distant brain regions and transient EEG signals). The control-diffusion mechanism in FEH ensures that hypergraph representations evolve smoothly over time , mimicking the gradual emergence of stable brain states.

## 4 Experiments

### 4.1 Data and Pre-processing

The fMRI-EEG dataset we used is LEMON, which is open access and was referenced in the literature [18], encompasses data from 227 healthy participants. This cohort includes a young group (N=153, age 25.1±3.1 years, range 20–35 years, 45 females) and an elderly group (N=74, age 67.6±4.7 years, range 59–77 years, 37 females). Participants underwent resting-state fMRI and a 62-channel EEG experiment. The preprocessing protocol for rs-fMRI data was conducted utilizing the Nipype framework as defined in [18]. Data preprocessing procedures for EEG were executed utilizing EEGLAB105 (version 14.2.1b) [19]. The preprocessing entailed the application of principal component analysis (PCA) for the purpose of dimensionality reduction. Participants also underwent the Test of Attentional Performance (TAP), which assessed their capacity for sustained attention; the Trail Making Test (TMT), which measures cognitive flexibility; the Vocabulary Test (WST), which indicates the measurement of verbal intelligence level and the assessment of language comprehension; and the LPS-2, which measures logical or inferential thinking and quantifies fluid intelligence. We divided the test results of the TAP, TMT, WST, and LPS-2 into two categories: high scoring group and low scoring group, which served as indicators to measure the effectiveness of the model. To enable comprehensive and equitable benchmarking against existing approaches, we introduce commonly used small-scale synchronized datasets named CN-EPFL dataset [20] with 20 individuals. Further details on datase tand Pre-processing can be found in Supplementary Material.

## 4.2 Implementation Details for Experiment

We validated the performance of our model by differentiating high scoring group and low scoring group. The 5-fold cross-validation (CV) scheme was used to obtain a reliable evaluation on the performance of competing models and the proposed FE-NET. Classification performance was assessed using multiple metrics, including classification accuracy (Acc), precision (Pre), and specificity (Spe).

For dynamic fMRI-EEG hypergraphs embedding module, the $\alpha_e$ and $\alpha_e$ is set as 0.05 and 0.9, the learning rate we set is $10^{-2}$, weight decay we set is $5 \times 10^{-4}$, the dropout we used is 0.2, we use 5 multi-layer Interactive Hyperedge Neurons modules to form the generator for fMRI -EEG hypergraphs, and the $\eta$ is we set is 0.6. The parameter settings and introductions of competing models are shown in Supplementary Material.

Table 1: Comparing FE-NET's performance (average accuracy±standard deviation) with other state-of-the-art methods

| TAP-WST | Method | Acc | Pre | Spe | TMT-LPS | Acc | Pre | Spe |
|---|---|---|---|---|---|---|---|---|
| | BrainGNN [21] | 67.23±2.15 | 71.89±6.42 | 70.59±3.17 | | 61.38±3.27 | 60.54±3.85 | 61.27±4.36 |
| | M-GAT-BC [22] | 70.19±4.82 | 70.57±5.03 | 65.39±4.21 | | 60.83±7.61 | 65.71±6.92 | 62.83±2.49 |
| High | SGP-SL [23] | 55.62±4.17 | 59.48±4.76 | 56.29±6.84 | High | 59.35±3.24 | 61.09±5.73 | 57.92±3.18 |
| Sustained | Cross-GNN [24] | 74.83±5.29 | 71.06±5.47 | 74.35±4.82 | Cognitive | 68.57±3.63 | 63.82±5.49 | 62.45±3.71 |
| Attention: | TAN [25]: | 73.12±5.37 | 70.49±3.85 | 71.84±4.93 | Flexibility | 68.29±6.83 | 66.71±5.03 | 63.62±3.47 |
| VS. | RH-BrainFS [26] | 74.26±5.13 | 71.93±4.58 | 73.74±3.81 | VS. | 61.29±3.51 | 66.38±4.27 | 66.54±5.18 |
| Low | MCRLN [27] | 65.71±4.53 | 70.28±2.94 | 69.83±6.21 | Low | 66.47±5.76 | 61.73±4.62 | 62.84±3.49 |
| Sustained | MMP-GCN [28] | 79.42±5.19 | 70.58±5.49 | 71.95±3.27 | Cognitive | 66.29±6.83 | 68.37±5.68 | 61.93±7.84 |
| Attention | FE-NET | **85.29±2.57** | **83.46±3.19** | **80.83±4.05** | Flexibility | **79.18±3.47** | **75.29±4.83** | **74.38±5.62** |
| | FE-NETnoFEH | 78.45±3.82 | 78.63±2.47 | 72.91±2.73 | | 73.57±3.64 | 69.27±4.18 | 66.84±3.59 |
| | FE-NETnoFED | 79.35±2.67 | 80.17±5.83 | 76.82±6.45 | | 70.46±8.53 | 67.38±5.29 | 67.49±5.84 |
| | BrainGNN [21] | 65.93±3.74 | 68.27±4.85 | 67.38±3.29 | | 60.84±4.57 | 61.29±3.87 | 60.93±4.18 |
| | M-GAT-BC [22] | 68.53±3.18 | 69.47±4.63 | 64.72±3.85 | | 62.38±4.19 | 64.59±5.47 | 61.73±2.94 |
| High | SGP-SL [23] | 56.84±4.73 | 60.38±4.19 | 58.62±5.84 | High | 59.47±3.29 | 61.83±5.29 | 57.46±3.85 |
| Verbal | Cross-GNN [24] | 72.59±5.47 | 69.84±5.18 | 73.27±4.73 | Logical | 68.29±3.58 | 63.47±5.39 | 62.84±3.29 |
| Intelligence | TAN [25]: | 71.38±4.85 | 67.29±3.18 | 69.47±4.29 | Intelligence | 68.53±6.47 | 66.82±5.84 | 63.19±3.47 |
| VS. | RH-BrainFS [26] | 73.62±4.18 | 68.53±4.73 | 72.84±3.85 | VS. | 61.29±3.47 | 66.38±4.29 | 66.47±5.18 |
| Low | MCRLN [27] | 64.82±4.73 | 69.47±2.85 | 68.29±6.47 | Low | 66.53±5.29 | 61.84±4.18 | 62.38±3.19 |
| Verbal | MMP-GCN [28] | 77.29±5.84 | 68.53±5.47 | 70.38±3.19 | Logical | 66.84±6.73 | 68.29±5.39 | 61.47±7.47 |
| Intelligence | FE-NET | **82.19±2.84** | **80.47±3.85** | **78.53±4.73** | Intelligence | **77.38±3.47** | **73.29±4.85** | **72.84±5.47** |
| | FE-NETnoFEH | 78.29±3.85 | 77.38±2.84 | 73.62±2.47 | | 73.47±3.29 | 69.53±4.18 | 66.29±3.85 |
| | FE-NETnoFED | 79.84±2.73 | 79.62±5.29 | 75.47±6.18 | | 70.53±8.47 | 67.38±5.84 | 67.29±5.47 |

## 4.3 Overall Evaluation

These competing methods and our FE-NET can be categorized into two groups: (Group A): fMRI or EEG modeling method including BrainGNN, M-GAT-BC, and SGP-SL. (Group B): Multimodal brain imaging modeling method including Cross-GNN, TAN, RH-BrainFS, MCRLN, and MMP-GCN.

As shown in Table 1, on one hand, the proposed FE-NET generally significantly outperforms three single-modal brain imaging modeling methods in four different tasks by a large margin, which substantiates the effectiveness of the fusion between fMRI and EEG.

On the other hand, compared with five multimodal brain imaging modeling methods, our FE-NET yields consistently better results in all metrics, Cross-GNN aimed at capturing inter-modal dependencies through dynamic graph learning and mutual learning mechanisms. Specifically, the inter-modal representations are carefully coupled into the combined space to infer the dependencies between modalities. However, Cross-GNN can not eliminate complex data distribution between fMRI and EEG. In contrast, our approach successfully narrow the gap between the hemodynamic responses and the neural oscillations through synchronization operation of GAN-based OFEI and IHEN.

TAN introduces a triple network to extract discriminative information from the high-order representation feature space obtained from multi-modal data. It incorporates self-attention to dynamically estimate the importance of brain regions and utilizes the cross-attention mechanism to extract complementary information from different modalities. However, TAN overlooks the absence of a one-to-one correspondence between fMRI and EEG and used static systems, resulting in the loss of complementary information between the two modalities. In contrast, our method depict the continuous dynamics of hidden representation by Neural ODEs, thereby capturing this relationship.

**Generalization capability evaluation:** To further evaluate our generalization capability, we conducted additional benchmarking on the synchronized CN-EPFL datasets. The result is shown in

Supplementary Material, all methods were trained on LEMON and tested on CN-EPFL, all models were both trained and tested on CN-EPFL. The results demonstrate that our model still exhibits significant advantages, whether generalizing to task-related datasets or directly utilizing synchronized datasets, further validating its superiority.

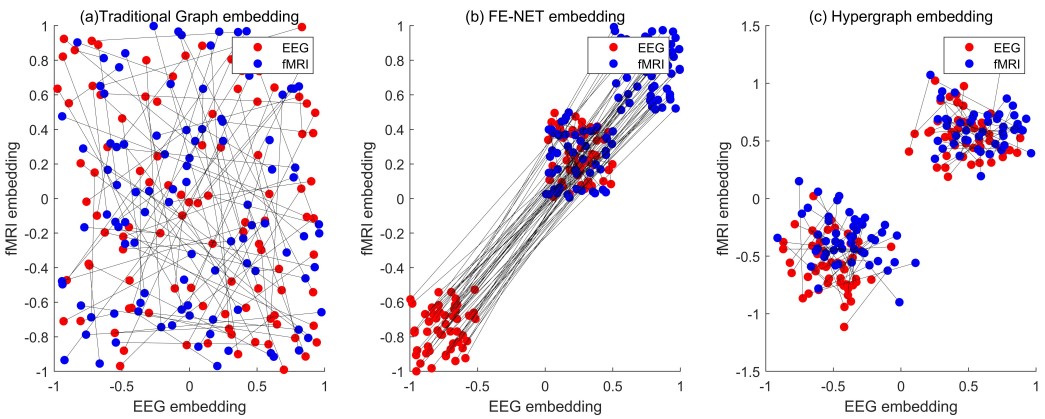

Figure 2: Visualization of the modality gap between the EEG and fMRI modalities.

## 4.4 FED, and FEH Modules Analysis

As shown in Table 1, we conducted ablation experiments on FED and FEH modules, FE-NETnoFEH and FE-NETnoFED are variants of FE-NET. FE-NETnoFEH employs initial hypergraphs generated by DHC algorithm as inputs for the FED module, whereas FE-NETnoFED utilizes a traditional hypergraph neural network for graph embedding. The superior performance of FE-NET compared to these two variants demonstrates the effectiveness of the modules we proposed. We have attempted to use traditional methods to model the process of transitioning from $x(t_k)$ to $x(t_{k+1})$. As shown in Supplementary Material, the performance of FE-NET (RNN, LSTM, GRU) is not as effective as FE-NET.

The ablation study of OFEI and IHEN in FEH is conducted by evaluating the average accuracy in four downstream tasks. The result is shown in the Supplementary Material. The variants of FED that exclusively employ OFEI or IHEN are referred to as "FEH(OFEI)" and "FEH(IHEN)", respectively. The performance of FEH outperforms "FEH(OFEI)" and "FEH(IHEN)", demonstrating its effectiveness. Additionally, the efficacy of the control and diffusion processes within the FED is validated. The variants of FED that employ exclusively control processes or diffusion processes are denoted as "FED(Con)" and "FED(Diff)" respectively. A comparative analysis of the two processes reveals that the inclusion of only the control process or diffusion process, which relies solely on initial vertex features and neglects the hypergraph structure, results in inferior outcomes. We provided an algorithm complexity and computation resources analysis of FEH-FED and comparison of computational resources with other SOTA methods in the Supplementary Material.

**Modality embedding space analysis:** As demonstrated in [29], a modality gap exists within multimodal learning, wherein information from different modalities is situated in entirely distinct embedding spaces. This modality gap is correlated with model performance.

To gain a deeper understanding of the performance improvements attributed to the proposed Neural ODEs-based hypergraph strategy, we present a visualization of the modality gap between the EEG and fMRI modalities within the LEMON dataset as shown in Figure 2. A comparison of FE-NET with the hypergraph and traditional graph embedding reveals that traditional graph embedding results in a larger modality gap, which in turn leads to decreased performance. Meanwhile, compared to hypergraph embedding, FE-NET enhances the clustering of embeddings between EEG and fMRI modalities in the embedding space. Notably, the embedding spaces of certain ROI in fMRI are relatively close to each other in Figure 2 (b) when compared to hypergraph embedding in Figure 2 (c). Multiple studies suggest that some ROI of fMRI show a strong correlation with EEG[30], whereas some show no significant association. These observations provide interpretability for the superior performance of Neural ODEs-based hypergraphs in FE-NET.

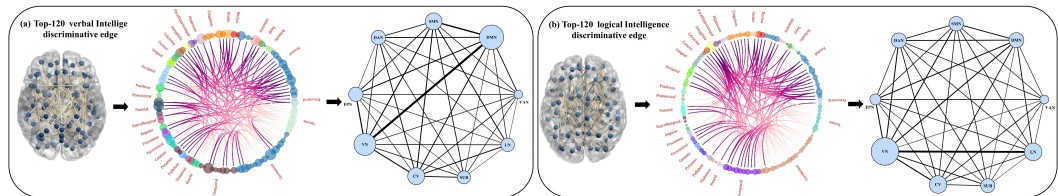

Figure 3: Functional connectivity of verbal intelligence and logical intelligence networks analysis

### 4.5 Functional connectivity networks analysis

To identify which FC are most discriminative, we use the method described in the literature [31] to rank the important Hyperedges. We summarize the top-120 verbal intelligence and logical intelligence discriminative FC and visualize them for analysis in Figure 3. The FC associated with verbal intelligence, as shown in Figure 3(a), exhibits the highest scores predominantly within the Default Mode Network (DMN), a fundamental FC network consistently identified in fMRI studies [32]. Within the DMN, the FCs demonstrating the strongest predictive capacity are primarily localized in the dorsal anterior cingulate cortex and the posterior superior temporal cortex. These regions, positioned at the intersection of the frontal and temporal lobes, constitute key components of the theory of mind network. In contrast, as shown in Figure 3(b), the FC most strongly associated with logical intelligence is primarily situated within the VN. Furthermore, the verbal intelligence-related FC in Figure 3(a) associated with the posterior superior temporal cortex has been ascertained to possess a reliable predictive value within the Ventral Attention Network (VAN) [33]. The FCs with the highest efficacy are located in the parahippocampal gyrus, and superior parietal lobule [34].

## 5 Conclusion

This paper designed a novel fMRI-EEG modeling method based on Neural ODEs named FE-NET. FE-NET performs substantially better than many state-of-the-art methods. The contributions of our research are as follows:

1) We propose a unified framework with two key components: FEH module and FED module. FEH can synthesize hypergraphs at multiple spatial and temporal scales, allowing the GAN to learn the joint distribution of hemodynamic and neural-oscillation signals and thus narrow the modality gap. FED, by contrast, uses Neural ODEs to capture latent temporal dependencies, inherently reconciling inconsistent sampling rates and temporal misalignment through learned differential equations governing asynchronous data dynamics.

2) For the first time, we attempt to model asynchronous fMRI and EEG data as Neural ODEs based hypergraph. Compared to previous studies that relied on tiny-sized synchronized datasets, the robustness and practicality of our model are further ensured.

Extensive comparative experiments and ablation studies validate the effectiveness of our proposed model and the corresponding two modules. Meanwhile, compared to simultaneously recorded fMRI-EEG data, asynchronously acquired single modality of EEG data is significantly less costly, which demonstrates the practical applicability of our method.

## 6 Acknowledgements

We would like to thank the reviewers and the chairs for their suggestions and efforts.

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

# A    Technical Appendices

All additional results can be downloaded and found in the supplementary material.

