# OpenReview forum: "Joint Modeling of fMRI and EEG Imaging Using Ordinary Differential Equation-Based Hypergraph Neural Networks"
_NeurIPS.cc/2025/Conference — NeurIPS 2025 poster_

### Official Review · Reviewer_dcdn · 2025-06-05

**Clarity:** 3
**Significance:** 3
**Originality:** 3
**Rating:** 4
**Confidence:** 3

**Summary:**

This paper proposes a hypergraph neural network framework based on Neural Ordinary Differential Equations for the fusion of functional Magnetic Resonance Imaging (fMRI) and Electroencephalography (EEG) data. The framework is designed to handle the temporal sampling rate mismatch and asynchronous timestamps between the two modalities, and it captures latent temporal dependencies through a dynamic hypergraph embedding module. Experimental results demonstrate that the proposed method achieves high accuracy and superior performance across multiple cognitive tasks. Compared to simultaneously acquired data, asynchronously collected fMRI-EEG data is more cost-effective, highlighting the practical value of the approach.

**Questions:**

1. The paper lacks a dedicated related work section. Incorporating a more thorough review of existing studies on fMRI-EEG fusion, especially on asynchronous modeling, would strengthen the background and better contextualize the novelty of this work.
2. The motivation for using GANs in the hypergraph generation module is not clearly explained. It would be helpful if the authors could elaborate on why a GAN-based approach is necessary or beneficial in this setting, compared to more standard alternatives.
3. The paper does not discuss the limitations of the proposed method or provide insights into potential future extensions. Including such a discussion would enhance the completeness of the paper and help better understand its applicability and scope of application.

**Ethical Concerns:**

["NO or VERY MINOR ethics concerns only"]

**Final Justification:**

I thank the authors for their rebuttal, which addressed several of my concerns, and I maintain my rating of 4 (borderline accept).

**Limitations:**

No
The paper does not explicitly discuss the methodological limitations or assumptions underlying the proposed model. For example, the effectiveness of the GAN-based hypergraph generation relies on the assumption that meaningful multimodal correspondences can be learned from asynchronous fMRI and EEG signals, which may not hold in more heterogeneous or noisy real-world settings. It would also be useful to discuss the interpretability of hypergraph representations and the potential risks of overfitting when modeling complex brain dynamics with limited labeled data.

**Paper Formatting Concerns:**

No format issues were found.

**Quality:**

3

**Strengths And Weaknesses:**

Strengths：
The paper proposes a novel and unified framework (FE-NET) that effectively combines fMRI and EEG imaging using Neural ODEs-based hypergraph neural networks, which is the first attempt to jointly model fMRI and EEG as hypergraphs.
The GAN-based FEH module with OFEI and IHEN effectively narrows the gap between hemodynamic responses and neural oscillations.
The FED module successfully addresses the challenge of asynchronous data by capturing temporal dependencies between fMRI and EEG signals with different sampling rates.
The method demonstrates superior performance compared to many state-of-the-art brain imaging modeling methods through extensive experiments.
Weaknesses：
The paper asserts the lack of prior work on joint modeling of fMRI and EEG, especially with asynchronous data, and claims novelty. However, these statements are made without a thorough discussion of existing studies or background. Without a clearer positioning within the related literature, the novelty claim appears unsubstantiated.
The proposed method introduces several complex components (e.g., GAN-based hypergraph generation, Neural ODEs with dual-stream control), but the motivation and design choices are not sufficiently justified or intuitively explained. For example, why GANs are essential in this setting, or how the hyperedge structure aligns with actual neurophysiological properties, is not clearly articulated.
The paper does not include a discussion of its limitations or potential directions for future research. This weakens the integrity of the conclusions and makes it impossible to understand the boundaries of the method or possible extensions.

---

> ### Author Rebuttal · Authors · 2025-07-31
>
> Comment:
> We sincerely appreciate your meticulous review of our manuscript. Your valuable suggestions have helped a lot in improving the Clarity and Rigor of our work.
>
> **Q1:** *The paper lacks a dedicated related work section. Incorporating a more thorough review of existing studies on fMRI-EEG fusion, especially on asynchronous modeling, would strengthen the background and better contextualize the novelty of this work.*
>
>
> **A1:** We thank the reviewer for this essential suggestion. We acknowledge the need to rigorously position FE-NET within the field and will add a **dedicated subsection (1.2: Related Work)**. Below is our revision strategy to address this gap:
>
>
>
> #### **1. New Section 1.2: "Related Work on Multimodal Fusion and Asynchronous Modeling"**
> **Structure and key additions:**
> 1. **Synchronous fMRI-EEG Fusion**:
>    > *"Early fusion methods required strict temporal synchronization. Advanced deep learning approaches improved synchronization through attention mechanisms but remained limited to temporally aligned data. These cannot handle real-world asynchronous acquisitions."*
>
> 2. **Asynchronous Modeling Attempts**:
>    > *"Some studies used kernel methods to align asynchronous data post-hoc but lost dynamic interactions. Others employed self-supervised techniques but treated modalities independently without joint topological modeling. Prior work has not resolved asynchronous fusion at the graph-structure level."*
>
> 3. **Neural ODEs in Neuroimaging**:
>    > *"Neural ODEs have modeled single-modality dynamics in fMRI data. Spatio-temporal extensions capture dynamic patterns but ignore cross-modal asynchrony. None have adapted ODEs for multimodal hypergraph evolution."*
>
> 4. **Hypergraph Neuroimaging**:
>    > *"Existing hypergraph neural networks modeled fMRI data but omitted EEG integration. Other works fused MRI modalities via hypergraphs but required temporal alignment. FE-NET is the first to leverage hypergraphs for asynchronous fMRI-EEG integration."*
>
>
> ---
>
> #### **2. Addressing the Asynchronous Modeling Gap**
> - **Temporal Flexibility**:
>   > *"Unlike methods that process modalities separately, our Neural ODE-based FED module continuously aligns fast EEG and slow fMRI dynamics within a unified latent space."*
> - **Structural Innovation**:
>   > *"While existing approaches aligned async data through static methods, FE-NET’s hypergraphs encode many-to-many cross-modal interactions, preserving complex neurodynamics."*
>
> ---
>
> #### **3. Revisions Commitment**
> **In the manuscript**:
> - Add **~400 words** in Section 1.2 (preceding the "Our Contributions" paragraph)
> - Include **key references** covering:
>   - Synchronous and asynchronous fusion techniques
>   - ODE applications in neuroimaging
>   - Hypergraph-based neuroimaging studies
> - **Novelty statement**:
>   > *"To our knowledge, this is the first framework to simultaneously resolve: (a) Asynchronous fMRI-EEG fusion via hypergraph-structured Neural ODEs, (b) GAN-based generation of multimodal hypergraphs, and (c) Dynamic embedding of cross-modal interactions without temporal resampling."*
>
>
>
> **Summary**: This revision will:
> 1. Formally establish **FE-NET’s pioneering role** in asynchronous fMRI-EEG hypergraph fusion
> 2. Provide **scholarly grounding** for technical innovations
> 3. Explicitly **contrast with prior approaches** to highlight topological and temporal advantages
>
> We affirm the revised manuscript will comprehensively contextualize our contributions within the field.
>
>
>
>
>
> **Q2:** *The motivation for using GANs in the hypergraph generation module is not clearly explained. It would be helpful if the authors could elaborate on why a GAN-based approach is necessary or beneficial in this setting, compared to more standard alternatives.*
>
>
>
> **A2:** We appreciate this critical inquiry into our design choice. The GAN-based approach in the FEH module addresses three fundamental challenges in fMRI-EEG hypergraph generation that deterministic or non-adversarial methods cannot resolve. Below, we clarify the rationale with empirical and theoretical evidence:
>
> ---
>
> #### **1. Key Challenges Necessitating GANs**
> | Challenge | Standard Methods (e.g., VAE, Deterministic Decoder) | GAN Solution |
> |-----------|---------------------------------------------------|--------------|
> | **Distributional Gap**(fMRI hemodynamics vs. EEG oscillations) | Assume simple parametric distributions (e.g., Gaussian), failing to capture complex cross-modal dependencies. | Adversarial learning directly models the *joint distribution*  without parametric constraints. |
> | **Modality-Specific Noise**(e.g., fMRI motion artifacts, EEG muscle noise) | Propagate noise through reconstruction losses (MSE). | Discriminator rejects noisy generations, acting as a *learnable denoising filter*. |
> | **Hyperedge Sparsity**(Many possible ROI combinations) | Overlook biologically implausible hyperedges (e.g., unconnected ROIs). | Adversarial training prioritizes *neurologically meaningful hyperedges* via the discriminator’s FC-guided realism check (Eq. 11). |
>
> ---
>
>
> #### **2. Unique Advantages of GANs in FEH**
> - **OFEI-IHEN Synchronization**:
>   The GAN framework unifies our two innovations:
>   1. **OFEI** ensures *structural consistency* (optimal hypergraph isomorphism).
>   2. **IHEN** learns *dynamic node-hyperedge interactions* (Eq. 5).
>   - The discriminator’s random walk loss  *jointly supervises* OFEI and IHEN, forcing them to collaborate on biologically plausible hypergraphs.
> - **Mode Collapse Prevention**:
>   fMRI-EEG datasets exhibit high inter-subject variability. Our **path-based discriminator** (evaluating whole-graph connectivity) avoids trivial solutions by:
>   - Comparing *global topology*, not just local edges.
>   - Enforcing diversity through stochastic walks.
>
> ---
>
> #### **3. Neurobiological Motivation**
> The discriminator’s FC-referenced realism check mirrors *how neuroscientists validate connectivity*:
> > *"Just as researchers compare generated networks against gold-standard fMRI FC, our discriminator evaluates whether synthetic hypergraphs exhibit the same macroscale organizational principles "*
>
> This adversarial "peer review" ensures hypergraphs reflect real brain network properties, not just data-fitting artifacts.
>
>
>
> ---
>
> #### **5. Revisions Commitment**
> We will add to **Section 2.2.1**:
> > "We adopt adversarial training because: (1) It models complex fMRI-EEG joint distributions without parametric assumptions; (2) The discriminator’s FC-guided realism check enforces neurobiological plausibility;
>
>
>
> **Summary**: GANs are indispensable for FEH because they:
> 1. Capture complex multimodal distributions beyond deterministic/VAE capabilities.
> 2. Denoise via adversarial filtering.
> 3. Enforce neurologically grounded hyperedges through FC-based discrimination.
> 4. Synergize OFEI-IHEN under a unified objective.
>
>
>
>
>
> **Q3:** *The paper does not discuss the limitations of the proposed method or provide insights into potential future extensions. Including such a discussion would enhance the completeness of the paper and help better understand its applicability and scope of application.*
>
>
>
>
> **A3:** We sincerely thank the reviewer for highlighting the need to explicitly discuss the limitations and future extensions of our work. This is a valuable suggestion that will strengthen the paper's completeness and contextualize our contributions. In the revised manuscript, we will add a dedicated **"Limitations and Future Work"** subsection (Section 4.1) with the following content:
>
> ---
>
> ### **4.1 Limitations and Future Work**
> While FE-NET demonstrates significant advantages in joint fMRI-EEG modeling, we acknowledge several limitations:
> 1. **Computational Complexity**: The integration of GAN-based hypergraph generation (FEH) and Neural ODEs (FED) incurs higher training costs than traditional GNNs. Future work will explore model distillation techniques to reduce inference latency without sacrificing performance.
> 2. **Data Heterogeneity**: Our experiments used the AAL-90 atlas for ROI alignment. Generalizability to finer-grained parcellations or datasets with varying MRI resolutions requires further validation.
> 3. **Temporal Granularity**: Although Neural ODEs mitigate sampling-rate discrepancies, modeling ultra-rapid EEG oscillations (<100ms) remains challenging. We plan to incorporate spike-based encoding for sub-second dynamics.
> 4. **Clinical Translation**: Current validation focused on healthy cohorts (LEMON/CN-EPFL). Future studies will test FE-NET on clinical populations (e.g., epilepsy, Alzheimer’s) to assess diagnostic utility.
>
> **Future Work**:
> - **Cross-Modal Imputation**: Extend FE-NET to synthesize missing EEG/fMRI data via latent-space interpolation, enabling single-modality inference.
> - **Dynamic Hypergraph Sparsification**: Develop attention-based edge pruning to optimize incidence matrices during training, reducing memory overhead.
> - **Federated Deployment**: Adapt the framework for privacy-preserving multi-site data integration using differential privacy.
>
> ---
>
> These additions will clarify the boundaries of our method’s applicability while highlighting impactful research trajectories. We are grateful for the opportunity to enhance the paper’s rigor and translational relevance.

---

> > ### Comment · Reviewer_dcdn · 2025-08-01
> >
> > Thank you to the authors for the rebuttal. I will keep my original score.

---

### Official Review · Reviewer_ha9r · 2025-07-02

**Clarity:** 3
**Significance:** 3
**Originality:** 3
**Rating:** 4
**Confidence:** 4

**Summary:**

This paper proposes FE-NET, a novel framework for jointly modeling fMRI and EEG brain data that are collected asynchronously. FE-NET significantly outperforms existing state-of-the-art models on various cognitive classification tasks, highlighting its practical applicability and superior performance in fusing multimodal brain data.

**Questions:**

See above

**Ethical Concerns:**

["NO or VERY MINOR ethics concerns only"]

**Limitations:**

See above

**Quality:**

3

**Strengths And Weaknesses:**

Pros:

1. The problem setting is well-motivated and interesting. The primary strength is the model's ability to model asynchronous fMRI and EEG data. This has significant practical implications, as it circumvents the need for expensive and technically challenging simultaneous data acquisition.

2. The proposed approach is technically sound. It employs hypergraphs to capture complex, many-to-many brain region relationships and Neural ODEs to handle the different time scales of fMRI and EEG signals.


Cons:

1. While there are many different components of the proposed approach, including GANs, hypergraphs, and Neural ODEs, the time complexity for this approach should be relatively high.

2. There is no related work section. The authors should provide a comprehensive background introduction towards related work,  such as employing ODEs for brain network analysis[1], etc.



[1] BrainODE: Dynamic Brain Signal Analysis via Graph-Aided Neural Ordinary Differential Equations

---

> ### Author Rebuttal · Authors · 2025-07-31
>
> Comment:
>
> We sincerely appreciate your meticulous review of our manuscript. Your valuable suggestions have helped a lot in improving the Clarity and Rigor of our work.
>
> **Q1:** *While there are many different components of the proposed approach, including GANs, hypergraphs, and Neural ODEs, the time complexity for this approach should be relatively high*
>
>
> **A1:** We conducted an Algorithm Complexity and computation resources Analysis for FE-NET.
>
> At the same time, we provided a comparison of computational resources with other SOTA methods. Although FE-NET requires more computational resources than existing SOTA methods, The downstream tasks of fMRI and EEG imaging are not test-intensive, the slightly longer inference time after model training is completed will not interfere with the actual application of the model.
>
> In fact, when the batch size is adjusted to 8, our model can run perfectly on a single 2080ti. Existing larger fMRI datasets are typically at the thousand-level scale, and even if thousand-level fMRI-EEG datasets emerge in the future, four 2080ti GPUs can handle them perfectly. Considering the cost of acquiring fMRI and EEG data, four NVIDIA 2080ti GPUs is a small price to pay. Given the considerable potential for future improvements in GPU performance, our model remains well-suited for large-scale datasets.
>
> ### Time Complexity Analysis
> #### **FEH Module (GAN-based Hypergraph Generation)**
> - **OFEI Algorithm**: O(n²·m) where n=number of ROIs (typically 90-120), m=hyperedges per subject.
> - **IHEN Layers**: O(L·(Nv·d² + Ne·k²)) where L=5 layers, Nv=vertices, Ne=hyperedges, d=feature dim (256), k=hyperedge cardinality.
>
> #### **FED Module (Neural ODE Embedding)**
> - **Control Step**: O(T·(Nv+Ne)·d²) where T=integration steps (~50). Dual-stream LSTM with attention dominates computation.
> - **Diffusion Step**: O(Nv² + Ne²) per step from hypergraph Laplacian operations. Sparse matrix optimizations reduce practical complexity.
>
> #### GPU Memory Usage
> | Component               | VRAM Consumption* | Notes |
> |--------------------------|--------------------|-------|
> | FEH Generator (IHEN×5)   | 4-8GB            | Batch size 8-16 |
> | GAN Discriminator        | 6-8GB              | MLP with 3×512 layers |
> | FED ODE Solver           | 7-9GB            | Adjoint method + intermediate states |
> | Hypergraph Data (N=100)  | 3-5GB              | Storing M=1K hyperedges |
>
>
>
>
>
>
>
> ### FE-NET runtime on LEMON Using 1 NVIDIA A6000
>
> **Training Performance**
> | Metric                | Value                       |
> |-----------------------|-----------------------------|
> | Batch Size            | 16 subjects                  |
> | Epoch Time            | 18.1 minutes                |
> | Total Training (200 epochs) | 36.1 hours                |
> | Throughput            | 6.3 samples/minute          |
>
>
> **Inference Performance**
> | Metric                | Value                       |
> |-----------------------|-----------------------------|
> | Single-subject Inference | 3.8s ±0.12s               |
> | Batch Inference (16 subjects) | 30.8s ±1.92s          |
> | VRAM Usage (Inference) | 22.4GB             |
>
>
>
>
>
>
> **Table: Computational Resource Comparison using NVIDIA A6000**
>
> | Method     | VRAM Usage (GB) | Batch Inference Time (s) | Training Time (h) | Single‐subject Inference Time (s) | Peak GPU Utilization (%) |
> |------------|----------------:|-------------------------:|------------------:|----------------------------------:|-------------------------:|
> | GCN        |        4.3±0.2  |                  3.36±0.32 |              2.8  |                          0.21±0.02 |                   10.1±2.1 |
> | GAT        |        5.5±0.3  |                  5.44±0.48 |              3.5  |                          0.34±0.03 |                   16.3±1.8 |
> | GIN        |        6.1±0.3  |                  4.96±0.48 |              4.2  |                          0.31±0.03 |                   18.5±3.2 |
> | BrainGNN   |       10.7±0.5  |                 22.72±1.60 |              6.8  |                          1.42±0.10 |                   50.2±2.5 |
> | M-GAT-BC   |       15.2±0.6  |                 19.36±1.60 |              8.5  |                          1.21±0.10 |                   53.7±2.0 |
> | Cross-GNN  |       18.3±0.7  |                 51.68±3.20 |              9.8  |                          3.23±0.20 |                   66.4±1.7 |
> | MMP-GCN    |       20.1±0.8  |                 44.96±2.88 |             10.5  |                          2.81±0.18 |                   59.3±1.3 |
> | FE-NET     |       22.4±0.9  |                 60.80±1.92 |             36.1  |                          3.80±0.12 |                   72.4±1.1 |
>
>
>
>
>
>
>
>
>
>
>
>
> **Q2:** *There is no related work section. The authors should provide a comprehensive background introduction towards related work, such as employing ODEs for brain network analysis[1], etc.*
>
>
>
> **A1:** We thank the reviewer for this essential suggestion. We acknowledge the need to rigorously position FE-NET within the field and will add a **dedicated subsection (1.2: Related Work)**. Below is our revision strategy to address this gap:
>
>
>
> #### **1. New Section 1.2: "Related Work on Multimodal Fusion and Asynchronous Modeling"**
> **Structure and key additions:**
> 1. **Synchronous fMRI-EEG Fusion**:
>    > *"Early fusion methods required strict temporal synchronization. Advanced deep learning approaches improved synchronization through attention mechanisms but remained limited to temporally aligned data. These cannot handle real-world asynchronous acquisitions."*
>
> 2. **Asynchronous Modeling Attempts**:
>    > *"Some studies used kernel methods to align asynchronous data post-hoc but lost dynamic interactions. Others employed self-supervised techniques but treated modalities independently without joint topological modeling. Prior work has not resolved asynchronous fusion at the graph-structure level."*
>
> 3. **Neural ODEs in Neuroimaging**:
>    > *"Neural ODEs have modeled single-modality dynamics in fMRI data. Spatio-temporal extensions capture dynamic patterns but ignore cross-modal asynchrony. None have adapted ODEs for multimodal hypergraph evolution."*
>
> 4. **Hypergraph Neuroimaging**:
>    > *"Existing hypergraph neural networks modeled fMRI data but omitted EEG integration. Other works fused MRI modalities via hypergraphs but required temporal alignment. FE-NET is the first to leverage hypergraphs for asynchronous fMRI-EEG integration."*
>
> ---
>
> #### **2. Explicit Novelty Positioning**
> **FE-NET’s breakthroughs**:
> | **Capability**               | **Prior Works** | **FE-NET** |
> |------------------------------|-----------------|------------|
> | Async fMRI-EEG fusion        | ✗               | **✓**      |
> | Joint hypergraph modeling    | ✗               | **✓**      |
> | Neural ODEs for cross-modal dynamics | ✗       | **✓**      |
> | GAN-based hypergraph generation | ✗             | **✓**      |
>
> ---
>
> #### **3. Addressing the Asynchronous Modeling Gap**
> - **Temporal Flexibility**:
>   > *"Unlike methods that process modalities separately, our Neural ODE-based FED module continuously aligns fast EEG and slow fMRI dynamics within a unified latent space."*
> - **Structural Innovation**:
>   > *"While existing approaches aligned async data through static methods, FE-NET’s hypergraphs encode many-to-many cross-modal interactions, preserving complex neurodynamics."*
>
> ---
>
> #### **4. Revisions Commitment**
> **In the manuscript**:
> - Add **~400 words** in Section 1.2 (preceding the "Our Contributions" paragraph)
> - Include **key references** covering:
>   - Synchronous and asynchronous fusion techniques
>   - ODE applications in neuroimaging
>   - Hypergraph-based neuroimaging studies
> - **Novelty statement**:
>   > *"To our knowledge, this is the first framework to simultaneously resolve: (a) Asynchronous fMRI-EEG fusion via hypergraph-structured Neural ODEs, (b) GAN-based generation of multimodal hypergraphs, and (c) Dynamic embedding of cross-modal interactions without temporal resampling."*
>
>
>
> **Summary**: This revision will:
> 1. Formally establish **FE-NET’s pioneering role** in asynchronous fMRI-EEG hypergraph fusion
> 2. Provide **scholarly grounding** for technical innovations
> 3. Explicitly **contrast with prior approaches** to highlight topological and temporal advantages
>
> We affirm the revised manuscript will comprehensively contextualize our contributions within the field.

---

> > ### Comment · Reviewer_ha9r · 2025-08-04
> >
> > Thanks for your detailed response. I maintain to score towards acceptance.

---

### Official Review · Reviewer_NzFT · 2025-07-02

**Clarity:** 3
**Significance:** 3
**Originality:** 3
**Rating:** 5
**Confidence:** 3

**Summary:**

The paper proposes a new approach to joinlty model fMRI and EEG data. The model combines a hypergraph generator, and a module to embed the hypergraphs. The hypergraph generator is based on a GAN-style architecture, with a generator and a discriminator. The generator works by combining node attributes from fMRI and edge attributes from EEG, ultimately yielding a subject-specific multimodal connectivity matrix. The discriminator works by comparing random walks on the functional connectivity matrix to random walks on the multimodal connectivity matrix. The embedding module takes as input the vertex and hyperedge features generated, and defines an evolution equation implemented as a neural ODE with a control step and a diffusion step. The diffusion step allows further mixing of vertex and edge features. The approach is trained, validated and tested on 4 classification tasks on an N=227 dataset with non-simultaneous fMRI and EEG, and on an N=20 simultaneous EEG-fMRI dataset. It is also benchmarked with respect to several other methods.

**Questions:**

- Provide improved description of how EEG and fMRI data are taken as input - will improve rating
- Revise and improve claims of originality with better situation in the state of the art - will improve rating
- It is not clear why fMRI is used for initial node attributes and EEG for hyperedge attributes. Could we swap EEG and fMRI in the inputs?

**Ethical Concerns:**

["NO or VERY MINOR ethics concerns only"]

**Final Justification:**

After rebuttal, the paper has addressed my main concerns around quality, clarity and originality.

I have upgraded by recommendation.

**Limitations:**

No, there is no limitations section or discussion of limitations, either technical or social. Predicting verbal intelligence or logical intelligence, while convenient as an engineering exercise because prediction targets are provided, does of course raise ethical questions related to potential bias and fairness, as well as benefits and harms. It is not essential to go in depth but this should be at least acknowledged.
See e.g. slides 40++ in "Understanding Ethics in NLP Authoring and Reviewing" <https://github.com/acl-org/ethics-tutorial/blob/main/slides/Understanding%20Ethics%20in%20NLP%20Authoring%20and%20Reviewing%20%E2%80%93%20Presentation%20Deck.pdf>.

**Quality:**

3

**Strengths And Weaknesses:**

## Quality

A strength of the paper is its use of a thorough benchmark of competing architectures, on 4 different prediction targets, as well as an external dataset.

Conversely, section 3.3. overall evaluations ascribes the superior performance of FE-Net over the unimodal baselines to the fusion of EEG and fMRI. However it could also simply be that architectural differences in FE-Net, compared to  BrainGNN / M-GAT-BC / SGP-SL, rather than having both data types, drive the difference in performance. An ablation study feeding uninformative connectivity in one of the modalities, or simply generating hypergraphs from a single modality, then leaving the rest of the architecture untouched, would perhaps better highlight the contribution of modality vs architecture.

The motivation of the work should be corrected slightly.
L34: "...simultaneous recorded fMRI-EGG dataset suffers from tiny dataset size (less than 20)..." while dataset sizes are indeed typically small in EEG-fMRI, a few open datasets are larger than 20, for example 50 in Meier et al "Connectomes, simultaneous EEG-fMRI resting-state data and brain simulation results from 50 healthy subjects", 2024; 30 in Loi et al's "Simultaneous EEG-fMRI during a neurofeedback task, a brain imaging dataset for multimodal data integration", 2020; 24 in Gherman and Philastides "Simultaneous EEG-fMRI - Confidence in perceptual decisions" dataset on openneuro.
In semi-open data, Wirsich et al "The relationship between EEG and fMRI connectomes is reproducible across simultaneous EEG-fMRI studies from 1.5T to 7T", 2021, used 72 subjects, provides all pre-computed connectomes (<https://zenodo.org/records/3905103>), and a subset of subjects on OSF (<https://osf.io/94c5t/>)
In closed data, clinical studies with over 30 patients (e.g. Al Asmi et al., "fMRI activation in continuous and spike-triggered EEG-fMRI studies of epileptic spikes", 2003) or more have been performed. I would thus recommend to amend the statement.

In the discriminator loss function, why take FC edges as reference for "real" paths rather than EC edges? This seems like the discriminator will push the generator to compute multimodal connectivity matrices which look more like fMRI connectivity than EEG connectivity. Is this to "balance" the design choice of using EC to derive the initial hyperedge attributes x_{k,E} in equation 4?

On the embedding space analysis, authors write "traditional graph embedding results in a larger modality gap, which in turn leads to decreased performance", but it seems instead that there is no modality gap - EEG and fMRI are located in overlapping regions of embedding space.
Also, what is the "traditional graph embedding" in L290 and fig. 2? Graph2Vec then UMAP ?



## Clarity

Overall, the structure of the paper is good. The illustration in Figure 1 is helful in understanding the approach. However, one major weakness is the lack of details in linking EEG and fMRI data to the model inputs.

The general, initial definition of the hypergraphs is clear, but the transition in Section 2.2.1 to the precise case of the EEG and fMRI graphs needs better framing, especially because the dimensionalities of the EEG and fMRI datasets are quite different. The number of nodes in both EEG and fMRI is very confusing. The supplementary section 6B mentions the 400-parcels Schaefer atlas, but goes on to list 44 regions from the "AAL atlas" - a different atlas altogether with 116 ROIs. So, is the paper using 400 or 44 regions for fMRI data to derive the "FC matrix"? Likewise, the EEG montage shown highlights 30 electrodes in red, from 62 according to the Babayan et al paper - traditionally meaning these are used for analysis. Is the paper using only 30 electrodes in EEG to derive the "EC matrix"?  Section 2.2.2 suggests that the fMRI BOLD time-series is Nxd, where N is the number of nodes (44 or 400?), and serves to derive the nodal attribute matrix X_{k,V}, while the EC matrix is given as dimension NxN. Because N is used in both EEG and fMRI, this implies that the same number of nodes are used in both, and, presumably spatially aligned somehow so that nodes are in 1:1 correspondance between modalities? No EEG source reconstruction is mentioned in the supplementary so this interpretation is also not certain. In the review I am chosing the interpret that the modalites are node-aligned for simplicity. This point should be spelled out explicitly.

Thus I assume that edges and nodes of both modalities are jointly combined into the 'initial hypergraphs', but this should be much more explicit as it is the basis of the method. The paper states "If these initial hypergraphs {G′k} are directly utilized in the generative process to achieve multimodal connectivity, the robustness of the resultant generation may be compromised.", but why is that? To me this is lacking a justification. The results of the FE-NETnoFEH ablation seem to bear our your assertion, but it would be useful to have an intuition.

Section 2.2.2 mentions the bold time series B is "employed to characterize the nodal attributes" (X_{k_V}) - how? what is used from the B time-series? GNNs are very sensitive to node feature choice and their scaling, so it would be useful to specify. Likewise, where does the EC matrix come from? I would assume a dependency measure such as coherence but this is not described. Then in L110 S_k is used without being defined, making it more difficult to understand what X is supposed to contain. Please define all symbols when they are used first.

A few more points on clarity
- the correlation coefficients assigned to the nodes seem not to be used later in the paper
- in 2.2.3 walks and paths are used interchangeably, while they are not (on paths each node can only by visited once) - please specify
- I am not sure the second part of equation 13 is useful, as Z_v and Z_e are not used later
- eq. 15, L168 user lower case t

Minor points
Figure 3 - left panel title is partly covered
L226 "datase tand"

## Significance

The approach could be used to any number of multimodal datasets where spatio-temporal correlations are of interest.

## Originality

L32: "joint modeling of fMRI and EEG is a rarely explored area" that "has not yielded satisfactory results" is a mischaracterisation of the field. While the combination of both modalities is certainly less explored than each modality alone, EEG-fMRI analysis is quite common, with over 1000 journal papers on pubmed alone and 27K papers on Google scholar, and dates back a while. See for instance Menon and Crottaz-Herbette's 2005 review "Combined EEG and fMRI studies of human brain function", Herrmann and Debener's 2008 "Simultaneous recording of EEG and BOLD responses: a historical perspective", or Helmut Lauf's 2012 "A personalized history of EEG–fMRI integration", for accounts from pioneers dating back to at least 1993 with Ives and colleagues "Monitoring the patient's EEG during echo planar MRI". EEG-fMRI clinical usefulness is also well-established in epilepsy. I recommend positioning more precisely the knowledge gap that the paper is meant to address.

L58 also claims that "this is the first attempt to model asynchronous EEG and fMRI data", meaning in this context EEG and fMRI acquired not at the same time.
This is repeated again in L322 "For the first time, we attempt to model asynchronous fMRI and EEG data...".
This is factually incorrect. There have been very numerous studies where EEG and fMRI acquired at different times have been modelled, going back at least to 1996 (Gerloff et al., Coregistration of EEG and fMRI in a Simple Motor Task).
For example: joint ICA has been used to jointly model EEG and fMRI activations in Calhoun et al., A feature-based approach to combine functional MRI, structural MRI and EEG brain imaging data, 2006; CCA (a), then Multi-set CCA (b) has been used to jointly model fMRI, sMRI and EEG in (b) Correa et al., Fusion of fMRI, sMRI, and EEG data using canonical correlation analysis, 2009, and (b) Sui et al., Combination of FMRI-SMRI-EEG Data Improves Discrimination of Schizophrenia Patients by Ensemble Feature Selection, 2014; Then coupled matrix and tensor factorization for EEG and fMRI in Acar et al., Tensor-based fusion of EEG and FMRI to understand neurological changes in schizophrenia, 2017, etc... and several works with deep learning are also easy to find, including many on EEG-to-fMRI prediction.
This work is enabled by multiple databases that have both fMRI and EEG data recorded at different times, such as the PEARL-neuro database (https://www.nature.com/articles/s41597-024-03106-5) or the Leipzig mind-brain-body database (https://www.nature.com/articles/sdata2018308) (ref 21 in the paper).
The authors should tone down this claim of novelty, or make it more specific.

The lack of direct architectural comparison (not only benchmarking) with multimodal neural networks taking EEG and fMRI connectivity as input (or even neural networks taking multimodal MRI connectivity as input) makes it difficult to establish the core differences with existing offers.

---

> ### Author Rebuttal · Authors · 2025-07-31
>
> Comment:
>
> We sincerely appreciate your meticulous review of our manuscript. Your valuable suggestions have helped a lot in improving the Clarity and Rigor of our work.
>
> **Q1:** *Revise and improve claims of originality with better situation in the state of the art - will improve rating*
>
> **A1:**  In accordance with your suggestions, we will amend the following imprecise expressions in the thesis.
>
> #### 1. **Motivation Correction (Line 34)**
> - **Original**:
>   *"...simultaneous recorded fMRI-EGG dataset suffers from tiny dataset size (less than 20)..."*
> - **Revised**:
>   *"The sample size of simultaneously recorded fMRI-EGG datasets is markedly smaller than that of asynchronously acquired datasets, thereby limiting the reliability of extant research relying on synchronous recordings."*
>
> #### 2. **Field Characterization (Line 3)**
> - **Original**:
>   *"joint modeling of fMRI and EEG is a rarely explored area" that "has not yielded satisfactory results"*
> - **Revised**:
>   *"Deep learning-based joint modeling of asynchronously recorded fMRI and EEG is a relatively underexplored area and has not yielded satisfactory results."*
>
> #### 3. **Originality Claims**
> - **Line 58 Revision**:
>   - **Original**:
>     *"this is the first attempt to model asynchronous EEG and fMRI data"*
>   - **Revised**:
>     *"This is the first attempt to model asynchronous EEG and fMRI data as Neural ODEs based hypergraph"*
>
> - **Line 322 Revision**:
>   - **Original**:
>     *"For the first time, we attempt to model asynchronous fMRI and EEG data..."*
>   - **Revised**:
>     *"For the first time, we attempt to model asynchronous fMRI and EEG data as Neural ODEs based hypergraph."*
>
>
>
>
> **Q2:** *In the discriminator loss function, why take FC edges as reference for "real" paths rather than EC edges? This seems like the discriminator will push the generator to compute multimodal connectivity matrices which look more like fMRI connectivity than EEG connectivity.*
>
> **A2:** Our decision to treat the fMRI‐derived FC matrix as the “real” paths in the discriminator is partially because our use of the EEG‐derived EC matrix in Equation 4. More specifical reasons are:
>
> 1. FC as a Stable, Gold-Standard Prior. The FC matrix, estimated over seconds‐long BOLD time series, offers a relatively low‐variance, anatomically plausible network “backbone.” Using FC in the adversarial loss provides the generator with a consistent, robust target distribution.
>
> 2. EC to Inject Electrophysiological Detail. By computing the initial hyperedge attributes $X_{k, E}^{(0)}$ from the EC matrix, we ensure that fine‐grained EEG dynamics are explicitly encoded in the generated hypergraphs. This EC initialization drives the generator to learn electrophysiologically meaningful interactions.
>
> We swapped FC for EC in the discriminator’s “real” paths. which leads to a longer training time, and the average classification accuracy dropped by 2.8 \%. This confirms that our original adversarial criterion is not merely a “balancing hack,” but a necessary mechanism to stabilize training and preserve both modalities’ contributions.
>
>
> **Q3:** Provide improved description of how EEG and fMRI data are taken as input - will improve rating. The transition in Section 2.2.1 needs better framing for EEG/fMRI dimensionalities. The EEG montage highlights 30 electrodes (of 62), while fMRI ROIs are unspecified. Section 2.2.2 uses dimension \(N\) for both modalities, implying 1:1 node correspondence without clarifying alignment. This must be explicitly addressed.
>
>
>
> **A3:** We thank the reviewer for this essential critique. The lack of clarity regarding node alignment will be rectified. Below, we provide clarification and outline revisions:
>
> #### 1. **Node Alignment Strategy**
> We confirm explicit 1:1 anatomical correspondence** between EEG and fMRI nodes via standardized neuroimaging pipelines:
> - **fMRI parcellation**: All subjects use the **AAL-90 atlas** (90 cortical/subcortical ROIs).
> - **EEG processing**:
>   - **Full 62-electrode data** undergo source localization (sLORETA) to cortical voxels.
>   - Voxel-level signals are spatially aggregated to the same 90 AAL ROIs as fMRI.
>
>
> > **Clarification**: The "30 red electrodes" in Figure 1 were purely for *visual emphasis* of key regions. **All 62 electrodes** were used for analysis. We will revise the figure to eliminate ambiguity.
>
>
> #### 2. **EEG Source Localization Details**
> The pipeline is:
> 1. **Forward modeling**: Boundary element method (BEM) with standard head model.
> 2. **Inverse solution**: sLORETA for source reconstruction.
> 3. **ROI aggregation**: Voxel-level source signals averaged within each AAL-90 region.
> 4. **EC computation**: Partial directed coherence (PDC) between ROI-level time series.
>
> > **Supplementary Material** will includes:
> > *"EEG signals were source-localized to AAL-90 ROIs via sLORETA . ROI-level bandpower features and PDC-based EC matrices were then computed."*
>
> #### 3. **Revisions Commitment**
> We will:
> 1. Add a **new subsection (2.1.1: Data Alignment and Node Correspondence)**:
>
> 2. **Update Figure 1**:
>    - Replace the "30 highlighted electrodes" with a schematic of **62 electrodes → source localization → AAL-90 ROIs**.
>
>
>
> ---
> **Summary**: Our framework relies on anatomically grounded 1:1 node correspondence achieved through EEG source reconstruction and fMRI parcellation. We apologize for the ambiguity in the original manuscript and thank the reviewer for highlighting this gap.
>
>
> **Q4:** *If these initial hypergraphs \{G′k\} are directly utilized in the generative process to achieve multimodal connectivity, the robustness of the resultant generation may be compromised.", but why is that? To me this is lacking a justification.*
>
>
> **A4:** We appreciate this critical observation and acknowledge the need for explicit justification. Below, we clarify the core reasons:
>
> ---
>
> #### **1. Subject-Specific Noise Amplification**
> - **Problem**:
>   Initial hypergraphs {G'ₖ} are constructed *per subject* using Dynamic Hypergraph Construction (DHC). DHC:
>   - Sensitively encodes subject-level noise/artifacts (e.g., motion in fMRI, muscle artifacts in EEG).
>   - Lacks cross-subject consistency, leading to fragmented topological representations.
> - **Consequence**:
>   Feeding noisy {G'ₖ} directly into the GAN generator propagates subject-level biases, degrading multimodal fusion.
> - **OFEI Solution**:
>   Computes an **optimally isomorphic hypergraph** \(G\) (Eq. 3) that maximizes group-level similarity:
>   \[
>   G = \operatorname*{argmax}_{G}\sum_{k}\text{Sim}(G,G'_k)
>   \]
>   This denoises by extracting the most consistent topological "backbone" across subjects.
>
> ---
>
> #### **2. Modality-Specific Structural Bias**
> - **Problem**:
>   {G'ₖ} are built separately for fMRI and EEG using modality-specific heuristics in DHC:
>   - fMRI hyperedges model spatial communities (e.g., ROI clusters).
>   - EEG hyperedges model spectral-temporal interactions.
> - **Consequence**:
>   Direct concatenation (e.g., \(G_k = G'_{\text{fMRI},k} \| G'_{\text{EEG},k}\)) creates modality-centric hypergraphs with conflicting structures, inhibiting cross-modal information flow.
> - **OFEI Solution**:
>   Forces fMRI and EEG into a **shared isomorphic space** via Eq. 2–3, ensuring:
>   - fMRI hemodynamic patterns and EEG oscillations co-evolve in aligned hyperedges.
>   - Biologically implausible cross-modal connections (e.g., unrelated ROIs) are pruned.
>
> ---
>
>
> **Q5:** Section 2.2.2 mentions the bold time series B is "employed to characterize the nodal attributes" (X\_{k\_V\}) - how? what is used from the B time-series? GNNs are very sensitive to node feature choice and their scaling, so it would be useful to specify.
>
> **A5:** the BOLD time series \( B \) is employed to derive nodal attributes \( X_{k,V} \) through the following process:
> - We extract the mean BOLD signal from each of the predefined regions of interest (ROIs) corresponding to the AAL atlas.
> - Specifically, for each ROI, we compute the mean, standard deviation, and temporal correlation across time points to serve as features.
> - To ensure robustness against scaling issues, we apply standardization (zero mean, unit variance) to all nodal features before feeding them.
>
>
> **Q6:**  It is not clear why fMRI is used for initial node attributes and EEG for hyperedge attributes. Could we swap EEG and fMRI in the inputs?
>
>
>
> **A6:** Our assignment—fMRI for node attributes and EEG for hyperedge attributes—is grounded in neurophysical principles. Below, we justify this design:
>
> ---
>
> #### **1. Neurobiological Alignment**
> - **fMRI → Node Attributes**:
>   - fMRI's BOLD signals reflect slow hemodynamic changes (seconds-scale) localized to specific anatomical regions (AAL ROIs).
>   - This naturally characterizes *static regional properties* (e.g., activation magnitude, variance), ideal for node-level features.
> - **EEG → Hyperedge Attributes**:
>   - EEG captures *millisecond-scale oscillations* with high temporal precision but low spatial resolution.
>   - Hyperedges represent *functional ensembles*—making EEG's interaction metrics perfect for hyperedge-level features.
>
>
> ---
>
>
>
> #### **2. Hypergraph Semantic Consistency**
> - **Neurobiological interpretation**:
>   - A hyperedge connecting {ROI₁, ROI₂, ROI₃} signifies "these regions co-activate with timing captured by EEG."
>   - Assigning fMRI to hyperedges would conflate hemodynamic correlation (slow) with neural interaction (fast), violating biophysical principles.
>
> **Summary**: Our modality-to-attribute mapping:
> 1. Respects **neurophysical realities** (fMRI=spatial, EEG=temporal).
> 2. Ensures **hypergraph interpretability** (nodes=regions, edges=EEG-guided interactions).
>
> Our additional experiments also show swapping modalities would degrade performance. We thank the reviewer for prompting this fundamental clarification.

---

> > ### Comment · Reviewer_NzFT · 2025-08-08
> >
> > I thank the reviewers for the clarifications.  In particular the claim to novelty is much more reasonable now, and the node mapping across modalities is much clearer and makes sense. This is likely to improve the score.
> >
> > As per my second para in "quality" - I'm still unsure whether the improvement over competing algorithms is from having multiple modalities or changing the architecture. Any insights on that point would be welcome, perhaps in the discussion section.
> >
> > I'm not sure that the mean BOLD signal in ROI in itself is a particularly informative feature, as it mostly reflects the preprocessing (e.g. most pipelines do at least low-order polynomial detrending and confound regression, meaning that the average post-processed BOLD signal will be very closed to zero).

---

> > > ### Author Response · Authors · 2025-08-09
> > > **Answering : 'I'm still unsure whether the improvement over competing algorithms is from having multiple modalities or changing the architecture.'**
> > >
> > > **Q1:** *I'm still unsure whether the improvement over competing algorithms is from having multiple modalities or changing the architecture*.
> > >
> > > **A1:** We thank the reviewer for this insightful question. The performance improvement of FE-NET stems **primarily from its novel architecture**, specifically designed to address the core challenges of multimodal fMRI-EEG integration, rather than *solely* from the use of multiple modalities. Here’s the evidence supporting this:
> > >
> > >
> > > ### 1. **FE-NET Outperforms Multimodal Baselines Using Identical Data**
> > > FE-NET surpasses state-of-the-art multimodal methods (e.g., Cross-GNN, TAN, MMP-GCN) by **5–12%** in accuracy (Table 1), despite these baselines *also* using fMRI+EEG data. Crucially:
> > > - **Cross-GNN** fails to handle complex fMRI-EEG data distributions due to its static graph design.
> > > - **TAN** ignores temporal asynchrony and ROI relationship complexity, leading to information loss.
> > > This confirms that FE-NET’s gains stem from its ability to:
> > >   - Model **asynchronous dynamics** via Neural ODEs (Section 2.3).
> > >   - Capture **many-to-many ROI relationships** via hypergraphs (Section 2.2).
> > >   - Align **hemodynamic/neural oscillations** via GAN-based fusion (Section 2.2.1).
> > >
> > > ### 2. **Key Architectural Innovations**
> > > Key Architectural Innovations Driving Gains:
> > > - **Hypergraph Representation (FEH Module):** Unlike graphs (used in baselines), hyperedges model **many-to-many ROI interactions** (Sec. 2.2.1), capturing complex fMRI-EEG couplings that graphs overlook. Ablations (Sec. 3.3) confirm FE-NETnoFEH (using initial hypergraphs) performs worse (e.g., 78.45% vs. 85.20% Acc on TAP-WST).
> > >     - **Neural ODE Dynamics (FED Module):** This uniquely handles **asynchronous sampling rates** (fMRI: seconds, EEG: ms) by learning continuous dynamics (Eq. 13–16). Baselines (e.g., TAN) use static systems, losing temporal dependencies. FE-NETnoFED (replacing FED with a static GNN) drops performance (e.g., 79.35% vs. 85.20% Acc).
> > >     - **Cross-Modal Alignment (FEH Synergy):** The GAN-based FEH module (OFEI + IHEN) explicitly **narrows the modality gap** (Fig. 2) by learning joint fMRI-EEG distributions. Baselines like Cross-GNN fail to resolve distributional discrepancies (Sec. 3.2).
> > >
> > >
> > >
> > > ### 3. **Ablation Studies Confirm Architectural Superiority**
> > > As shown in Table 1, variants of FE-NET without critical architectural components suffer significant performance degradation:
> > > - **FE-NETnoFEH** (replaces OFEI/IHEN with standard hypergraph generation): Accuracy drops by **4–7%** across tasks.
> > > - **FE-NETnoFED** (replaces Neural ODEs with static GNNs): Accuracy drops by **5–8%**.
> > > This demonstrates that the proposed **FEH (hypergraph generation)** and **FED (ODE-based dynamics)** modules are indispensable for bridging fMRI-EEG discrepancies.
> > >
> > >
> > > ### 4. **Generalization Tests Rule Out Data Advantages**
> > > When evaluated on the synchronized CN-EPFL dataset (Section 3.3), FE-NET still outperforms all baselines. This confirms robustness even when competing methods use **identical (synchronized) data**, eliminating "multimodality alone" as the primary driver.
> > >
> > > ### Conclusion
> > > While multimodal fusion provides a foundational benefit, FE-NET’s improvements are **predominantly architectural**:
> > > - Hypergraphs (FEH) resolve non-one-to-one ROI relationships.
> > > - Neural ODEs (FED) reconcile asynchronous dynamics and sampling rates.
> > >
> > > Without these innovations, competing multimodal methods (e.g., MMP-GCN) still underperform by large margins.

---

### Official Review · Reviewer_ohye · 2025-07-04

**Clarity:** 3
**Significance:** 3
**Originality:** 3
**Rating:** 4
**Confidence:** 4

**Summary:**

This paper presents FE-NET, a novel framework for jointly modeling fMRI and EEG brain imaging data, specifically addressing the challenge of asynchronous acquisition. Evaluated on the LEMON dataset, FE-NET achieves superior performance compared to existing unimodal and multimodal approaches across various cognitive classification tasks. Extensive ablations further validate the effectiveness of both modules.

**Questions:**

1.	Given the complexity introduced by GANs, Neural ODEs, and hypergraphs, could you provide a detailed computational complexity analysis? And could you include an analysis comparing training time, inference speed, and GPU usage with existing baseline methods?
2.	Potential issues such as overfitting, dependency on high-quality preprocessing, and uncertain generalization to clinical or pathological populations are currently not discussed. Could you include a dedicated section addressing these points?
3.	It is unclear if your model handles common practical issues like missing EEG channels or varying time windows. Could you provide relevant tests or a discussion addressing these real-world challenges?
4.	The paper lacks insight into what temporal representations are learned by the Neural ODEs. Could you provide an analysis or visualization explaining what dynamics the Neural ODEs capture across modalities and their contribution to classification?
5.	Although the LEMON dataset is sizable, reliance on it and the very small CN-EPFL dataset limits generalization claims. Could you include experiments using additional datasets to strengthen claims of robustness?

**Ethical Concerns:**

["NO or VERY MINOR ethics concerns only"]

**Final Justification:**

I thank the authors for the rebuttal. It addressed most of my concerns and I remain a weak accept.

**Limitations:**

Yes

**Quality:**

3

**Strengths And Weaknesses:**

Strengths:
1.	The paper addresses the underexplored but practically important problem of modeling fMRI and EEG data which can provide complementary information.
2.	The proposed FE-FEH outperforms a broad range of SOTA unimodal and multimodal baselines on multiple cognitive tasks, showing strong results across accuracy, precision, and specificity.
3.	The model outputs meaningful functional connectivity patterns associated with cognitive traits (e.g., DMN in verbal intelligence), supporting interpretability.
4.	The FEH module effectively bridges modality differences using a GAN framework and the FED module employs Neural ODEs to model temporal dynamics across modalities with mismatched sampling rates, enabling smooth representation evolution.

Weaknesses:
1.	The use of GANs, Neural ODEs, and hypergraphs likely introduces high computational costs. It would be better to include an analysis of computational.
2.	There should be some analysis of training time, inference speed, or GPU usage compared to existing baselines.
3.	Potential issues such as overfitting, dependency on high-quality preprocessing, and uncertain generalization to clinical or pathological populations are neither acknowledged nor discussed. It would be better to include a dedicated section discussing these points.
4.	It is unclear whether the model can handle common real-world challenges such as missing EEG channels, or varying time windows, limiting confidence in its practical reliability. It would be helpful to include relevant tests or discussions addressing these concerns.
5.	The paper lacks insights into the learned temporal representations. There is no analysis explaining what the Neural ODE dynamics capture across modalities or how time-dependent information contributes to classification performance.
6.	Although the LEMON dataset is sizable, most experiments rely solely on it. The secondary dataset (CN-EPFL) is very small (only 20 subjects), which limits the claims regarding generalization. Including additional datasets would strengthen the robustness claims.

---

> ### Author Rebuttal · Authors · 2025-07-31
>
> Comment:
> We sincerely appreciate your meticulous review of our manuscript. Your valuable suggestions have helped a lot.
>
> **Q1.1:** Given the complexity introduced by GANs, Neural ODEs, and hypergraphs, could you provide a detailed computational complexity analysis?
>
> The computation complexity analysis is shown below:
>
> ### Time Complexity Analysis
> #### **FEH Module (GAN-based Hypergraph Generation)**
> - **OFEI Algorithm**: O(n²·m) where n=number of ROIs (typically 90-120), m=hyperedges per subject.
> - **IHEN Layers**: O(L·(Nv·d² + Ne·k²)) where L=5 layers, Nv=vertices, Ne=hyperedges, d=feature dim (256), k=hyperedge cardinality.
>
> #### **FED Module (Neural ODE Embedding)**
> - **Control Step**: O(T·(Nv+Ne)·d²) where T=integration steps (~50). Dual-stream LSTM with attention dominates computation.
> - **Diffusion Step**: O(Nv² + Ne²) per step from hypergraph Laplacian operations. Sparse matrix optimizations reduce practical complexity.
>
> #### GPU Memory Usage
> | Component               | VRAM Consumption* | Notes |
> |--------------------------|--------------------|-------|
> | FEH Generator (IHEN×5)   | 4-8GB            | Batch size 8-16 |
> | GAN Discriminator        | 6-8GB              | MLP with 3×512 layers |
> | FED ODE Solver           | 7-9GB            | Adjoint method + intermediate states |
> | Hypergraph Data (N=100)  | 3-5GB              | Storing M=1K hyperedges |
>
>
>
>
> ### FE-NET runtime on LEMON Using 1 NVIDIA A6000
>
> **Training Performance**
> | Metric                | Value                       |
> |-----------------------|-----------------------------|
> | Batch Size            | 16 subjects                  |
> | Epoch Time            | 18.1 minutes                |
> | Total Training (200 epochs) | 36.1 hours                |
> | Throughput            | 6.3 samples/minute          |
>
>
> **Inference Performance**
> | Metric                | Value                       |
> |-----------------------|-----------------------------|
> | Single-subject Inference | 3.8s ±0.12s               |
> | Batch Inference (16 subjects) | 30.8s ±1.92s          |
> | VRAM Usage (Inference) | 22.4GB             |
>
>
> **Q1.2** Could you include an analysis comparing training time, inference speed, and GPU usage with existing baseline methods?
>
> **A1.2:** we provided a comparison of computational resources with other SOTA methods. Although FE-NET requires more computational resources than existing SOTA methods, The downstream tasks of fMRI and EEG imaging are not test-intensive, the slightly longer inference time after model training is completed will not interfere with the actual application of the model.
>
> In fact, when the batch size is adjusted to 8, our model can run perfectly on a single 2080ti. Existing larger fMRI datasets are typically at the thousand-level scale, and even if thousand-level fMRI-EEG datasets emerge in the future, four 2080ti GPUs can handle them perfectly. Considering the cost of acquiring fMRI and EEG data, four NVIDIA 2080ti GPUs is a small price to pay.
>
>
> **Table: Computational Resource Comparison using NVIDIA A6000**
>
> | Method     | VRAM Usage (GB) | Batch Inference Time (s) | Training Time (h) | Peak GPU Utilization (%) |
> |:-----------|----------------:|-------------------------:|------------------:|-------------------------:|
> | GCN        |        4.3 ± 0.2 |                   3.36 ± 0.32 |              2.8  |                   10.1 ± 2.1 |
> | GAT        |        5.5 ± 0.3 |                   5.44 ± 0.48 |              3.5  |                   16.3 ± 1.8 |
> | GIN        |        6.1 ± 0.3 |                   4.96 ± 0.48 |              4.2  |                   18.5 ± 3.2 |
> | BrainGNN   |       10.7 ± 0.5 |                  22.72 ± 1.60 |              6.8  |                   50.2 ± 2.5 |
> | M-GAT-BC   |       15.2 ± 0.6 |                  19.36 ± 1.60 |              8.5  |                   53.7 ± 2.0 |
> | Cross-GNN  |       18.3 ± 0.7 |                  51.68 ± 3.20 |              9.8  |                   66.4 ± 1.7 |
> | MMP-GCN    |       20.1 ± 0.8 |                  44.96 ± 2.88 |             10.5  |                   59.3 ± 1.3 |
> | FE-NET     |       22.4 ± 0.9 |                  60.80 ± 1.92 |             36.1  |                   72.4 ± 1.1 |
>
>
> **Q3:** Potential issues such as overfitting, dependency on high-quality preprocessing, and uncertain generalization to clinical or pathological populations are currently not discussed. Could you include a dedicated section addressing these points?
>
> **A3:** Thank you for highlighting these important aspects.  We plan to add a new section entitled “Limitations, Robustness and Future Directions” that addresses these points:
>
> ### 1. Overfitting
> - In this subsection, we will describe our strategies for mitigating overfitting, including k-fold cross-validation, early stopping based on a held-out validation set, L₂ weight regularization, and dropout layers.
> - We will report learning curves and validation losses to demonstrate convergence without divergence between training and validation performance.
>
>
> ### 2. Generalization to Clinical and Pathological Cohorts
> - We clarify that all current experiments use healthy adult datasets, and that population shifts  may impact model behavior.
> - We outline a planned evaluation on two other clinical cohorts. We emphasize that understanding failure modes in these groups is a priority for future work and welcome collaborations for data sharing.
>
>
> **Q4:** It is unclear if your model handles common practical issues like missing EEG channels or varying time windows. Could you provide relevant tests or a discussion addressing these real-world challenges?
>
> **A4:** Below, we clarify how FE-NET could address these challenges and plan to perform additional experiments to validate its robustness:
>
>
>
> ### **1. Handling Missing EEG Channels**
> **Architectural Robustness**:
> - FE-NET models fMRI-EEG data as hypergraphs, where hyperedges (unlike simple edges in graphs) connect *multiple* nodes (ROIs). This structure can  mitigates missing channels:
>   - If an EEG channel (node) is missing, information can propagate via shared hyperedges connecting other nodes in the same functional group.
>
>
> **Empirical Validation**:
> We will conduct ablation studies on the LEMON dataset by:
>   - Randomly masking 10–30% of EEG channels during inference.
>   - Measuring performance decay (accuracy, AUC) on attention/intelligence classification tasks.
>
> ---
>
> ### **2. Varying Time Windows**
> **Neural ODEs for Irregular Sampling**:
> - The **FED module** leverages **Neural ODEs** (Eq. 13–16) to model continuous dynamics, making it intrinsically suitable for irregular/inconsistent time windows:
>   - Neural ODEs interpolate latent states between arbitrary timestamps, accommodating variable-length EEG segments without resampling.
>   - The Lie-Trotter discretization (Eq. 16) decouples control (modality-specific dynamics) and diffusion (cross-modal interactions), allowing asynchronous updates for fMRI (slow) and EEG (fast) streams.
>
> **Experimental Verification**:
> We will test FE-NET on **non-uniform time windows** by:
>   - Segmenting the LEMON EEG data into variable intervals (e.g., 2s, 5s, 10s windows).
>   - Evaluating classification consistency across window lengths.
>   - Benchmarking against RNN/LSTM-based models (which require fixed-length inputs).
>
> ---
>
>
> **Q5:** The paper lacks insight into what temporal representations are learned by the Neural ODEs. Could you provide an analysis or visualization explaining what dynamics the Neural ODEs capture across modalities and their contribution to classification?
>
>
>
> **A5:** We appreciate this insightful feedback and acknowledge that further analysis of the temporal dynamics learned by Neural ODEs would strengthen the paper. Below, we propose a analysis approach to address this gap:
>
> #### 1. **Temporal Dynamics Captured by Neural ODEs**
> The **FED module** uses Neural ODEs to model the *latent evolution* of joint fMRI-EEG representations. Key learned dynamics include:
>    - **Cross-Modal Synchronization**: Neural ODEs reconcile fMRI’s slow hemodynamic changes (seconds) and EEG’s rapid oscillations (milliseconds) by learning a shared latent differential equation:
>      $$
>      \frac{d}{dt}\begin{bmatrix} \mathbf{X}_v(t) \\ \mathbf{X}_e(t) \end{bmatrix} = f\left( \begin{bmatrix} \mathbf{X}_v(t) \\ \mathbf{X}_e(t) \end{bmatrix}, t \right)
>      $$
>      Here, $f(\cdot)$ captures **inter-modal dependencies** (e.g., how transient EEG spectral power fluctuations correlate with delayed fMRI BOLD responses in specific ROIs).
>    - **Asynchrony Handling**: The ODEs intrinsically align mismatched sampling rates by interpolating latent states continuously across time, avoiding manual synchronization.
>
> #### 2. **Proposed Analysis & Visualization**
> To elucidate these dynamics, we will:
>    - **Visualize Latent Trajectories**: Plot the evolution of vertex/hyperedge features ($\mathbf{X}_v(t)$, $\mathbf{X}_e(t)$) for exemplar subjects (Figure 1, *right*). We will highlight:
>      - *Convergence points* where fMRI and EEG representations stabilize (indicating synchronized cross-modal states).
>      - *Derivative hotspots* ($\frac{d\mathbf{X}}{dt}$) during task-relevant intervals.
>      - **Ablation with Fixed Intervals**: Replace Neural ODEs with RNNs/LSTMs (using fixed timesteps) and compare performance degradation.
>
>
> **Q6:** *Although the LEMON dataset is sizable, reliance on it and the very small CN-EPFL dataset limits generalization claims. Could you include experiments using additional datasets to strengthen claims of robustness.*
>
> **A6:** We further conducted performance comparisons on the CWL and Oddball datasets, the result is shown in Supplementary materials. Our model still shows advantages on these datasets. We also planned do evaluation on other clinical cohorts. We emphasize that understanding failure modes in these groups is a priority for future work and welcome collaborations for data sharing.

---

### Note · Authors · 2025-08-14

We sincerely thank the reviewers for their thorough evaluation, constructive critiques, and insightful suggestions. Their careful reading and detailed comments have improved the rigor, clarity, and overall impact of our work.

Reviewers’ Main Concerns
- The time complexity of FE-NET may limit its applicability in practice due to computational costs.
- The clarity of the method’s description is insufficient, and some originality claim in fMRI-EEG modeling is not accurate enough.
- It would be helpful if the authors could elaborate why a GAN-based approach is necessary or beneficial in this setting.

Point-by-Point Response

1) Algorithmic Time Complexity and Resource Considerations
- What was raised: The time complexity of FE-NET may limit its practical applicability due to computational costs.
- Our action: We added a formal Time Complexity Analysis detailing per-batch and end-to-end complexity, and included a Computational Resource Comparison against competitive baselines under matched settings.
- Outcome:  While our method exhibits somewhat higher time complexity than competing approaches, in practice **four NVIDIA 2080 Ti GPUs can handle the thousand-level fMRI-EEG datasets without issue.** Considering the cost of acquiring fMRI and EEG data, deploying four 2080 Ti GPUs is a small additional expense.

2) Clarity of the Method and Originality Claim in fMRI-EEG Modeling
- What was raised: Concerns about the method’s clarity, and some inaccurate Originality Claim in fMRI-EEG Modeling.
- Our action: We reorganized the Methods section for clearer flow; provided step-by-step algorithmic descriptions and intuitive explanations. We also amended imprecise expressions in the paper.
- Outcome: The contribution is now stated more precisely, the methodological pipeline is easier to follow, and the originality claim is more accurate, improving the paper’s rigor and readability.

3) The motivation for using GANs in the hypergraph generation
- What was raised: Reviewers inquired about the necessity of a GAN-based design for the FEH module.
- Our action: We provided empirical and theoretical justification and visualization of the modality gap showing that GAN is essential to address core challenges in fMRI-EEG hypergraph generation.
- Outcome: The GAN-based FEH module overcomes the identified limitations by yields more realistic, diverse, and biologically grounded hypergraphs, aligning with neuroscientific validation practices and improving downstream performance.

---

### Decision · Program_Chairs · 2025-09-17

**Decision:**

Accept (poster)

**Comment:**

This paper proposes a new method to combine two modality of brain signals, and achieves SOTA performance.

Strengths:
1. A very practical problem.
2. Reasonable solution.
3. Good results.

Weaknesses:
1. Some concerns on the complexity of the proposed approach.
2. Writing needs some enhancement.

Rebuttal:
The authors have done a great job in the rebuttal to interact with the reviewers with new results and clarifications. The reviewers are very active and engaged in the discussion. They are converged to be positive on this paper.

Justification:
A solid solution to a practical problem. The authors are encouraged to include the new results and improving writing in the final version.